# SAGE: Spatial-visual Adaptive Graph Exploration for Efficient Visual Place Recognition

**Shunpeng Chen[1], Changwei Wang[2], Rongtao Xu[3], Xingtian Pei[1], Yukun Song[1]**
**Jinzhou Lin[1], Wenhao Xu[1], Jingyi Zhang[1], Li Guo[1], Shibiao Xu[1]***

[1] School of Artificial Intelligence, Beijing University of Posts and Telecommunications
[2] Key Laboratory of Computing Power Network and Information Security, Ministry of Education,
Shandong Computer Science Center, Qilu University of Technology [3] Spatialtemporal AI
`shunpengchen@bupt.edu.cn`, `shibiaoxu@bupt.edu.cn`

## Abstract

Visual Place Recognition (VPR) requires robust retrieval of geotagged images despite large appearance, viewpoint, and environmental variation. Prior methods focus on descriptor fine-tuning or fixed sampling strategies yet neglect the dynamic interplay between spatial context and visual similarity during training. We present SAGE (Spatial-visual Adaptive Graph Exploration), a unified training pipeline that enhances granular spatial-visual discrimination by jointly improving local feature aggregation, organize samples during training, and hard sample mining. We introduce a lightweight Soft Probing module that learns residual weights from training data for patch descriptors before bilinear aggregation, boosting distinctive local cues. During training we reconstruct an online geo-visual graph that fuses geographic proximity and current visual similarity so that candidate neighborhoods reflect the evolving embedding landscape. To concentrate learning on the most informative place neighborhoods, we seed clusters from high-affinity anchors and iteratively expand them with a greedy weighted clique expansion sampler. Implemented with a frozen DINOv2 backbone and parameter-efficient fine-tuning, SAGE achieves SOTA across eight benchmarks. Notably, our method obtains 100% Recall@10 on SPED only using 4096D global descriptors. The code and model are available at `https://github.com/chenshunpeng/SAGE`.

## 1 Introduction

Visual Place Recognition (VPR) matches a query image to its corresponding location within a large-scale geotagged database, serving as a fundamental capability for critical applications such as autonomous robot navigation (Han et al., 2025), loop closure detection for autonomous driving (Teng et al., 2026), and large-scale map construction (Zhu et al., 2024). The main challenge of VPR is maintaining robust retrieval performance under severe and unconstrained environmental changes, including extreme viewpoint shifts, illumination variations, adverse weather, long-term temporal drift, and frequent dynamic occluders, among others (Liu et al., 2024; Zhu et al., 2025).

Early VPR methods relied on hand-crafted local descriptors (Lowe, 2004; Bay et al., 2008) and aggregated them into global encodings via pooling schemes such as Bag of Words or VLAD (Angeli et al., 2008; Jégou et al., 2010). However, these methods lack adaptability and perform poorly under large-scale appearance changes. With the advent of deep learning, learnable aggregation modules (Arandjelovic et al., 2016; Radenović et al., 2018) were introduced, which enhanced descriptor compactness and robustness by learning task-specific pooling strategies. Subsequent research has mainly focused on simplifying, regularizing, or refining aggregation mechanisms to improve generalization and computational efficiency (Jin et al., 2025b), such as reducing reliance on explicit cluster centers or alleviating the "burstiness" problem of local features (Lu et al., 2024d; Khaliq et al., 2024).

Recently, the advent of Visual Foundation Models (VFMs) (Dosovitskiy et al., 2020; Oquab et al., 2023) has advanced VPR by enabling the capture of long-range semantic dependencies and richer

---

*Corresponding author.

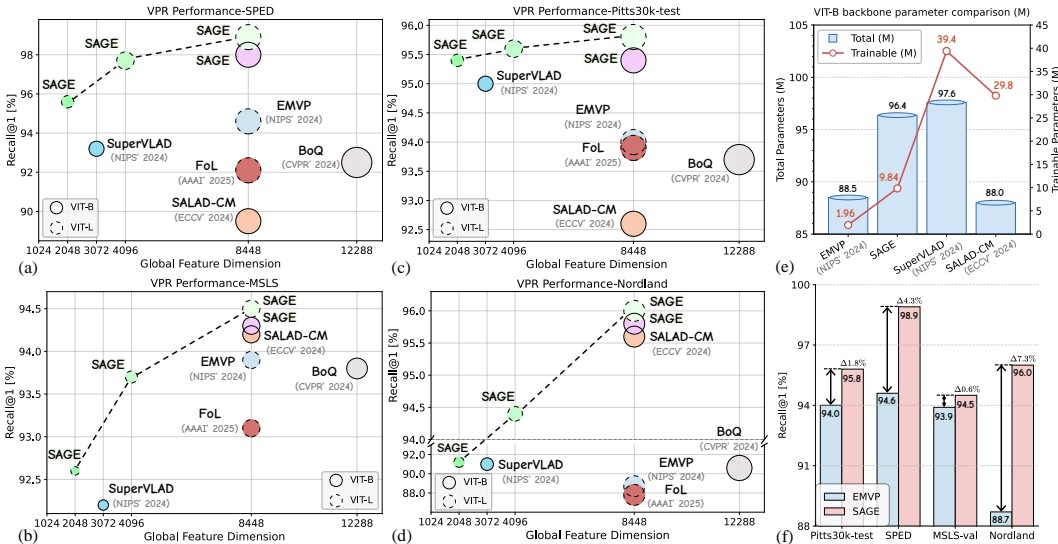

Figure 1: Performance and parameter efficiency of SAGE. (a–d) Recall@1 across four datasets at different global descriptor dimensions; SAGE achieves the best performance regardless of backbone and descriptor size. (e) Parameter comparison. By freezing DINOv2, SAGE substantially reduces **trainable** parameters compared to methods employing adapters or partial encoder tuning methods, demonstrating high efficiency. (f) Recall@1 performance compared with EMVP across the datasets.

interactions between image patches (Ali-bey et al., 2024; Lu et al., 2025a). These strategies reduce sensitivity to occlusion and background clutter and improve robustness with controlled parameter overhead. While recent adaptation methods for VFMs are notably efficient, different strategies within the broader VFMs ecosystem vary in resource consumption (Jin et al., 2025a). For instance, fine-tuning a backbone's encoder layers typically requires more computational resources than Parameter-Efficient Fine-Tuning (PEFT) approaches (Jia et al., 2022; Qiu et al., 2024). Furthermore, while the model's understanding is dynamic, a pre-defined or static sampling policy (Izquierdo & Civera, 2025) may fail to consistently present the most informative examples as learning progresses.

Recent research has explored constructing training batches that reflect real-world difficulty. However, a common strategy operates on a static "think-once, act-always" principle, relying on offline computations like pre-defined clustering based on initial features (Liao & Shao, 2022; Leyva-Vallina et al., 2023; Tzachor et al., 2025; Wang et al., 2025; Izquierdo & Civera, 2025). Such approaches often treat geographic priors and visual similarity independently, overlooking a crucial fact: what constitutes a real-world hard sample is governed by the intricate, dynamic interplay between spatial proximity and visual ambiguity. This hardness is not a fixed property, but a state that evolves with the model's embedding geometry during training. Effective mining thus requires an architectural "slow thinking" involving an iterative reassessment of difficulty. Without this, static methods gradually become obsolete. They continue feeding the model stale examples, wasting computational resources and limiting learning efficiency, as old challenges turn trivial and new ones emerge at the decision boundary. Ultimately, the inability to concurrently enhance granular local expressiveness and dynamically align the sampling process with the evolving geo-visual landscape leaves a significant gap in current training paradigms, potentially limiting their adaptability in real-world scenarios.

To address these interconnected limitations, we propose SAGE (Spatial-Visual Adaptive Graph Exploration), a unified VPR training framework that embraces a "slow thinking" paradigm for hard sample mining. Rather than depending on a one-time, fixed policy that labels samples as hard for the entire training run, SAGE continuously revisits and updates the hardness labels in response to changes in the model's representation. This philosophy is realized through a fundamentally dynamic architecture. At its core, an online process reconstructs a geo-visual affinity graph each epoch, ensuring the sampling strategy stays synchronized with the model's evolving embedding space. To maximize the impact of this intelligent sampling, SAGE also incorporates lightweight modules, including Soft Probing (SoftP) and an InteractHead, which enhance descriptor quality by amplifying discriminative local patches and modeling cross-image associations. This synergy between dynamic mining and enhanced feature representation allows SAGE to focus learning on the most informa-

tive spatial-visual neighborhoods, leading to state-of-the-art accuracy with remarkable parameter efficiency, as summarized in Fig. 1. In summary, the main contributions of this paper are:

- **SoftP Feature Interaction.** We propose SoftP, a lightweight module that uses data-driven residual weighting to enhance discriminative local patches, and an InteractHead that models associations between fragments across images, thereby improving descriptor coherence across views.

- **Dynamic Geo-Visual Graph Mining.** Our online strategy dynamically rebuilds the geo-visual affinity graph each epoch, keeping the mining process aligned with the model's evolving embedding space while prioritizing the most informative samples for faster convergence.

- **Weighted Greedy Clique Expansion.** Our weight-guided algorithm initiates sampling from anchors with high affinity and iteratively expands the most challenging neighborhoods, thereby generating balanced batches of utility that focus learning on detailed spatial and visual distinctions.

- **Efficient SOTA Accuracy.** Implemented with a frozen DINOv2 backbone and parameter-efficient fine-tuning, SAGE sets a new SOTA across eight challenging VPR benchmarks (Fig. 1), retaining robust retrieval competitiveness even with highly compact descriptors for large-scale deployment.

## 2    RELATED WORK

Visual Place Recognition (VPR) requires global descriptors to remain compact and robust under substantial variations in viewpoint, illumination, and scene structure. Early methods for generating global descriptors, such as NetVLAD (Arandjelovic et al., 2016), utilized a vast set of cluster centers, which rendered them vulnerable to domain shifts. Although self-supervised hard sample mining (Ge et al., 2020) has been introduced to mitigate this issue, such methods remain constrained by the representational capacity of CNN backbones, yielding suboptimal performance. With the incorporation of spatial attention (Noh et al., 2017) and feature reweighting (Ng et al., 2020), descriptor robustness has been yet improved. Other works focus on optimizing feature aggregation (Ali-Bey et al., 2023b; Izquierdo & Civera, 2024; Ali-bey et al., 2024), which improves performance but relies on fixed aggregation strategies and thus lacks adaptability to dynamically evolving embeddings. Recent studies have highlighted the effectiveness of modulating feature magnitudes in a lightweight, data-dependent manner prior to their aggregation. Such approaches include non-local attention for adaptive spatial weighting (Chen et al., 2023), efficient context encoding (Huang et al., 2022), parameter-efficient tuning with second-order moments (Gao et al., 2023), and the exponentially weighted fusion of pooling kernels (Stergiou & Poppe, 2022). Other methods reduce overhead by lowering the number of clusters or entirely eliminating clustering through centroid-free probes (CFP) (Lu et al., 2024d; Qiu et al., 2024), yielding compact descriptors by utilizing second-order feature statistics. Two-stage approaches (Chen et al., 2026; Lu et al., 2024a) improve retrieval accuracy through local feature re-ranking but introduce additional computational overhead. Recent single-stage methods can achieve comparable or even superior performance using only global features (Berton & Masone, 2025; Liu et al., 2025b). This pursuit of efficiency is also reflected in adapting powerful Visual Foundation Models (VFMs) for downstream tasks (Zhang et al., 2026; Liu et al., 2025a). With the proliferation of VFMs, Parameter-Efficient Fine-Tuning (PEFT) has emerged as a crucial paradigm. Instead of fine-tuning the entire backbone, these methods update only a small subset of parameters, such as lightweight adapters or normalization layers, thereby significantly enhancing training efficiency (Jia et al., 2022; Qiu et al., 2024). Our method follows this paradigm by freezing the backbone while introducing lightweight modules to enhance feature discriminability. Cross-image correlation methods (Lu et al., 2024b; Qiu et al., 2024) further enhance matching by capturing and modeling inter-image dependencies. In line with this paradigm, our approach strategically enhances feature discriminability. We introduce Soft Probing, a lightweight residual module that adaptively amplifies salient local regions, and InteractHead, which models cross-image dependencies. Together, they significantly boost descriptor quality with minimal parameter overhead.

In deep metric learning, dynamic sampling strategies (Liang et al., 2021) adjust the importance of training pairs with epoch-dependent weighting terms, organizing them in an "easy-to-hard" order, which enables the network to first learn general category boundaries from easy samples and then focus on hard samples in later stages. More recently, a graph-based sampling method (Liao & Shao, 2022) has been proposed, which constructs a nearest neighbor graph from class embeddings at the beginning of each epoch. By selecting an anchor class and its neighboring classes to form training batches, this approach improves the discriminative power of learned embeddings and enhances

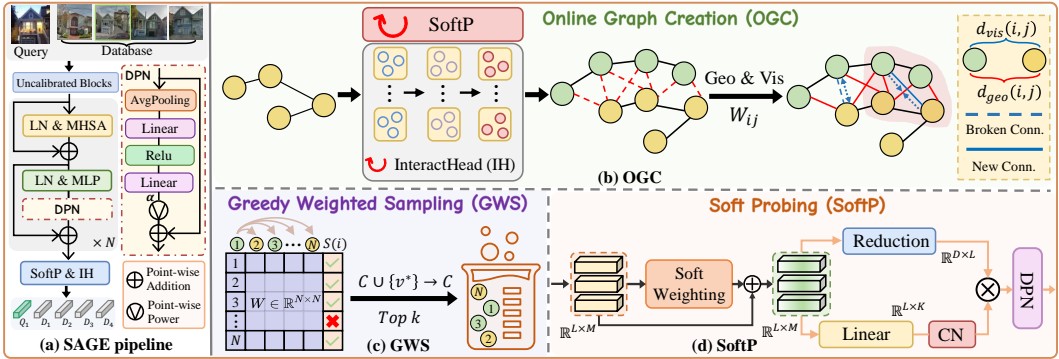

Figure 2: SAGE overview. (a) Pipeline: a frozen DINOv2 with PEFT outputs tokens; SoftP amplifies informative patches, and InteractHead applies cross-image attention to form a robust global descriptor. (b) Online Graph Creation: each epoch builds a geo–visual affinity graph, keeping top-k neighbors and updating edges as embeddings evolve. (c) Greedy Weighted Sampling: seed by average affinity and expand cliques by adding the most connected nodes. (d) SoftP: A lightweight module that uses residual weighting to emphasize discriminative features prior to aggregation.

training efficiency. In VPR, spatial graphs have been used to encode geographic relationships. For example, the MMS-VPR benchmark (Ou et al., 2025) represents street intersections and road segments as nodes and edges, leveraging topological context to improve retrieval performance. Such approaches often rely on mining discriminative regions or focusing on hard positive samples (Lu et al., 2023; Seidenschwarz et al., 2021; Fang et al., 2022; Wang et al., 2025) to enhance accuracy and robustness, but ignore geo-information. Other works reformulate VPR as a classification task (Berton et al., 2022a; 2023) to avoid explicit mining, but these remain limited by static sample selection and the neglect of geographic information. Moreover, on sparse datasets such as GSV-Cities (Ali-bey et al., 2022), these limitations restrict generalization capability. On dense datasets such as MSLS (Warburg et al., 2020), offline clustering methods (Izquierdo & Civera, 2025) partition visually similar and geographically neighboring images into fixed clusters for training. Another category of approaches leverages hybrid strategies (Kalantidis et al., 2020), hard negative mining (Garg et al., 2022; Deuser et al., 2023; Ali-Bey et al., 2023a) or generation (Peng et al., 2024) to enhance retrieval performance. However, most of these approaches struggle to generalize effectively. In contrast to such static or scheduled strategies, SAGE reconstructs a geo-visual graph at every training epoch and employs greedy sampling to focus on the most densely populated and challenging clusters in the evolving embedding spaces, achieving superior performance across datasets.

## 3 METHOD

Fig. 2 illustrates the proposed SAGE framework. First, the frozen DINOv2 feature extraction backbone processes input images (Sec.3.1). Next, the Soft Probing module aggregates these features into robust global descriptors (Sec.3.2). Then, Online Graph Creation employs InteractHead to refine descriptors and integrates geographic and visual distances to form dynamic graphs (Sec.3.3). Finally, we adopt greedy weighted sampling to focus training on hard examples (Sec.3.4).

### 3.1 FEATURE EXTRACTION

DINOv2 provides strong visual representations from large-scale self-supervised pretraining. We use a pretrained DINOv2 as a frozen backbone and achieve parameter-efficient fine-tuning by inserting learnable Dynamic Power Normalization (DPN) layers into the last $N$ encoder blocks (Oquab et al., 2023; Qiu et al., 2024). As shown in Fig. 2(a), an input image $I \in \mathbb{R}^{H \times W \times 3}$ passes through two stages. Uncalibrated Blocks extract base features, and Recalibrated Blocks integrate DPN to recalibrate features and produce task-specific representations. The backbone outputs one learnable class token and $L$ patch tokens, forming a token matrix $\mathbf{f} \in \mathbb{R}^{(L+1) \times M}$, where $M$ is the embedding dimension. SoftP and InteractHead then aggregate and refine these tokens to strengthen cross-image correspondence and yield a discriminative global descriptor $\mathbf{F} \in \mathbb{R}^{D \times K}$ for place recognition.

## 3.2 SOFT PROBING

Centroid-Free Probing (CFP) crafts a global image descriptor $\mathbf{f}$ by aggregating the $L$ local spatial descriptors $\{X_i\}_{i=1}^{L}$ into a second-order Gram-like covariance matrix, which eliminates the need for offline cluster centers (Qiu et al., 2024; Lu et al., 2024d). Although related methods like Moment Probing also leverage second-order statistics (Wang et al., 2022; Gao et al., 2023), a key limitation of CFP is its uniform treatment of all descriptors. This equal weighting can underemphasize subtle yet discriminative local cues. To address this shortcoming, we introduce Soft Probing (SoftP), a lightweight module shown in Fig. 2(d), which adaptively emphasizes informative spatial locations while preserving the underlying feature geometry before the final aggregation step.

SoftP first computes a scalar response for each descriptor and converts it into a bounded residual coefficient; this coefficient is broadcast across channels and applied in residual form to obtain the modulated descriptors. Concretely, for each descriptor $X_i$ we compute an $\ell_2$ response $s_i = \|X_i\|_2 + \varepsilon$ (with $\varepsilon > 0$ for numerical stability), and feed $s_i$ into a compact predictor $\phi$ (a two-layer MLP) that outputs a scalar which is squashed by a sigmoid and scaled by a hyperparameter $\alpha$:

$$\beta_i \;=\; \alpha \cdot \sigma\big(\phi(s_i)\big), \qquad 0 \le \beta_i \le \alpha. \tag{1}$$

The modulated descriptor is formed residually as:

$$\widetilde{X}_i \;=\; X_i + \beta_i X_i \;=\; (1+\beta_i)X_i. \tag{2}$$

This residual reweighting behaves like a soft, data-driven attention mechanism: it amplifies salient responses while avoiding destructive rescaling of channel structure (Ng et al., 2020). Under mild assumptions (or to first order when $\beta_i$ is small and the mean shift is negligible) (Gao et al., 2023), the variance of each dimension of the modulated descriptors increases approximately as:

$$\mathrm{Var}(\{\widetilde{X}_i\}) = \frac{1}{N}\sum_{i=1}^{N}(1+2\beta_i)\left\|X_i - \bar{X}\right\|^2 = \underbrace{\frac{1}{N}\sum_{i=1}^{N}\left\|X_i - \bar{X}\right\|^2}_{\mathrm{Var}(\{X_i\})} + \frac{2}{N}\sum_{i=1}^{N}\beta_i\left\|X_i - \bar{X}\right\|^2. \tag{3}$$

where $\bar{X} = \frac{1}{N}\sum_i X_i$. This relation highlights that SoftP selectively enlarges the variance contribution of high-response locations, thereby enhancing the sensitivity of subsequent aggregation stages to discriminative local structures. Finally, the set of modulated descriptors $\{\widetilde{X}_i\}$ is passed to the aggregation stage to produce the final global descriptor. SoftP adds only a negligible number of parameters, preserves the semantic geometry of the original descriptors, and consistently improves the robustness of the resulting global descriptor under significant appearance changes.

## 3.3 ONLINE GRAPH CREATION

To prepare for the graph creation, the image descriptors $\mathbf{f}_i \in \mathbb{R}^{D \times K}$ produced by SoftP are first processed by the InteractHead. Departing from prior method that partition features via learned cluster assignments(Lu et al., 2024b;d), InteractHead deterministically splits each descriptor into $S$ fixed length segments, $\{\mathbf{f}_i^{(s)}\}_{s=1}^{S}$. To enable cross-image attention, these segments are rearranged such that for each index $s$, the segments from all $B$ images form a sequence. These sequences are then processed by a two-layer Transformer encoder ($\mathcal{E}$) with GELU activation. This encoder structure applies attention across the batch for each segment type, capturing consistent correlations across views and improving descriptor robustness. The enhanced descriptors $\mathbf{F}$ are obtained by:

$$\mathbf{F} = \mathrm{reshape}\Big(\mathcal{E}\big([\mathbf{f}_1^{(1)}, \ldots, \mathbf{f}_B^{(1)}; \ldots; \mathbf{f}_1^{(S)}, \ldots, \mathbf{f}_B^{(S)}]\big)\Big), \tag{4}$$

Our approach continuously aligns candidate graphs to the model's current embedding space by reconstructing the graph at each training epoch. First, for each city we group images by their unique cluster labels and randomly sample one image from each cluster. The sampled images are passed through our model to obtain descriptors that serve as representative features for the clusters, producing a set of cluster-level features for every city.

The process begins by sampling cities with a probability proportional to their cluster count. From a chosen city, we randomly select a single cluster, termed a "place". We then identify $P$ similar

places by computing cosine distances between the descriptor of our selected place and all other cluster descriptors, and sampling probabilistically such that smaller distances yield a higher selection probability. The images from these $P + 1$ total places become the unordered nodes of a graph, for which we compute all pairwise Euclidean geographic distances $d_{\text{geo}}(i, j)$. Subsequently, we construct an adjacency graph by connecting nodes whose geographic distance is below a threshold $\tau$. From this graph, we extract several cliques (i.e., complete subgraphs), denoted as $G = (V, E)$. Finally, within each clique, we calculate the pairwise visual descriptor distances as $d_{\text{vis}}(i, j) = \|\mathbf{F}_i - \mathbf{F}_j\|_2$. To combine these two distance types and do so multiplicatively, we define:

$$W_{ij} = - \left( d_{\text{geo}}(i, j) \cdot d_{\text{vis}}(i, j) \right), \quad W_{ii} = 0. \tag{5}$$

As illustrated in Fig. 2(b), we construct a sparse affinity graph $\mathcal{G} = (V, E')$, where an edge $(i, j)$ exists if its affinity score $W_{ij}$ exceeds a predefined threshold $\tau_2$. This graph is dynamic, as it is rebuilt each epoch to reflect the continuous evolution of the model's embeddings. From this graph, we seek a complete subgraph, known as a clique, for the sampling process. The search concludes upon finding the first clique $C$ that meets a minimum size requirement of $|V_C| \geq N = 10$. This clique is then used to guide the subsequent sampling stage.

## 3.4 Greedy Weighted Sampling

We propose a greedy, weight-driven selection process that adaptively focuses training on the most informative neighborhoods while fully leveraging the reconstructed geo-visual graph. We first identify the most central node in the graph to serve as a cluster anchor. This is achieved by computing a seed score $S(i)$ for each node, which represents its total affinity to all other nodes:

$$S(i) = \frac{1}{N-1} \sum_{\substack{j=0 \\ j \neq i}}^{N-1} W_{ij}. \tag{6}$$

Fig. 2(c) shows the node with the highest score is chosen as the initial member of our training clique, $C = \{v_0^*\}$, where $v_0^* = \arg\max_i S(i)$. Subsequently, we iteratively expand the clique $C$ by adding the node $v^*$ that exhibits the highest average affinity to the current members of $C$:

$$v^* = \arg\max_{v \notin C} \frac{1}{|C|} \sum_{u \in C} W_{u,v}. \tag{7}$$

This procedure is repeated until the clique reaches the desired size $|C| = k$, where $k = 4$. By seeding from a central anchor and greedily expanding towards the closest nodes, our method effectively drills down into the densest subgraphs of the geo-visual landscape. These dense regions represent clusters of mutually confusing samples the most difficult scenarios where the model struggles to make fine-grained distinctions. Our approach is inherently adaptive to the model's learning progress. It dynamically responds to the evolving weight distribution each epoch, concentrating training effort on the most pertinent hard positive and negative examples. This adaptive focus not only accelerates convergence but also enhances the model's robustness against subtle spatial and visual ambiguities.

# 4 Experiments

## 4.1 Datasets and Performance Evaluation

We validate SAGE on a diverse collection of VPR benchmarks (Tab. 1) covering common real-world challenges: **Pitts30k-test** and **Pitts250k-test** (large viewpoint variation) (Torii et al., 2013), **SPED** (low-quality / high scene depth and condition changes) (Chen et al., 2017), **MSLS-val** (multi-year urban/suburban variability) (Warburg et al., 2020), **Nordland** (four-season natural scenes) (Sünderhauf et al., 2013), **Tokyo247** (multi-view urban captures) (Torii et al., 2015), **AmsterTime** (historical grayscale vs. contemporary RGB) (Yildiz et al., 2022), and **Eynsham** (rural grayscale route) (Cummins & Newman, 2010; Berton et al., 2022b). Further details can be found in App. A.3.

The experiment adopts Recall@N (R@N) as the evaluation metric, i.e., the percentage of query images for which at least one of the top-N retrieved database images geographically matches the query image (within a preset threshold). Thresholds follow standard protocols: 25 meters for Pitts30k-test, Pitts250k-test, Tokyo24/7, Eynsham and SPED; MSLS-val uses 25 meters with azimuth within 40 degrees; Nordland uses $\pm 10$ frames (Torii et al., 2013; Chen et al., 2017; Warburg et al., 2020).

Table 2: Comparison to SoTA Methods on VPR Benchmark Datasets. The best and second best metrics are shown in **red bold** and **blue bold**, respectively. Two-stage methods are denoted by [†].

| Method | Dim | SPED | | | Pitts30k-test | | | MSLS-val | | | Nordland | | |
|---|---|---|---|---|---|---|---|---|---|---|---|---|---|
| | | R@1 | R@5 | R@10 | R@1 | R@5 | R@10 | R@1 | R@5 | R@10 | R@1 | R@5 | R@10 |
| NetVLAD CVPR'2016 | 32768 | 70.2 | 84.5 | 89.5 | 81.9 | 91.2 | 93.7 | 53.1 | 66.5 | 71.1 | 6.4 | 10.1 | 12.5 |
| SFRS ECCV'2020 | 4096 | 80.2 | 92.6 | 95.4 | 89.4 | 94.7 | 95.9 | 69.2 | 80.3 | 83.1 | 16.1 | 23.9 | 28.4 |
| CosPlace CVPR'2022 | 512 | 75.5 | 87.0 | 89.6 | 88.4 | 94.5 | 95.7 | 82.8 | 89.7 | 92.0 | 58.5 | 73.7 | 79.4 |
| MixVPR WACV'2023 | 4096 | 84.7 | 92.3 | 94.4 | 91.5 | 95.5 | 96.3 | 88.0 | 92.7 | 94.6 | 76.2 | 86.9 | 90.3 |
| R2Former CVPR'2023 [†] | / | 67.5 | 75.8 | 77.8 | 91.1 | 95.2 | 96.3 | 89.7 | 95.0 | 96.2 | 77.0 | 89.0 | 91.9 |
| EigenPlaces ICCV'2023 | 2048 | 70.2 | 83.5 | 87.5 | 92.5 | 96.8 | 97.6 | 89.1 | 93.8 | 95.0 | 71.2 | 83.8 | 88.1 |
| SelaVPR ICLR'2024 | 1024 | 83.5 | 92.6 | 94.6 | 90.2 | 96.1 | 97.1 | 87.7 | 95.8 | 96.6 | 72.3 | 89.4 | 94.4 |
| SelaVPR ICLR'2024 [†] | / | 88.6 | 95.1 | 97.2 | 92.8 | 96.8 | 97.7 | 90.8 | 96.4 | 97.2 | 87.3 | 93.8 | 95.6 |
| CricaVPR CVPR'2024 | 4096 | 91.3 | 95.2 | 96.2 | 94.9 | 97.3 | 98.2 | 90.0 | 95.4 | 96.4 | 90.7 | 96.3 | 97.6 |
| SALAD CVPR'2024 | 8448 | 92.1 | 96.2 | 96.5 | 92.5 | 96.4 | 97.5 | 92.2 | 96.4 | 97.0 | 89.7 | 95.5 | 97.0 |
| EDTformer TCSVT'2025 | 4096 | 92.4 | 95.9 | 96.9 | 93.4 | 97.0 | 97.9 | 92.0 | 96.6 | 97.2 | 88.3 | 95.3 | 97.0 |
| BoQ CVPR'2024 | 12288 | 92.5 | 95.9 | 96.7 | 93.7 | 97.1 | 97.9 | 93.8 | 96.8 | 97.0 | 90.6 | 96.0 | 97.5 |
| SALAD-CM ECCV'2024 | 8448 | 89.5 | 94.9 | 96.1 | 92.6 | 96.8 | 97.8 | **94.2** | 97.2 | 97.4 | **95.6** | **98.6** | 99.1 |
| SuperVLAD NIPS'2024 | 3072 | 93.2 | 97.0 | 98.0 | 95.0 | 97.4 | 98.2 | 92.2 | 96.6 | 97.4 | 91.0 | 96.4 | 97.7 |
| EMVP NIPS'2024 | 8448 | 94.6 | 97.5 | 98.4 | 94.0 | 97.5 | 98.2 | 93.9 | **97.3** | 97.6 | 88.7 | 97.3 | **99.3** |
| FoL AAAI'2025 | 8448 | 92.1 | 96.5 | 98.0 | 93.9 | 97.2 | 98.1 | 93.1 | 96.9 | 97.4 | 87.8 | 94.5 | 96.4 |
| FoL AAAI'2025 [†] | / | 92.6 | 96.5 | 97.4 | 94.5 | 97.4 | 98.2 | 93.5 | 96.9 | 97.6 | 92.6 | 96.7 | 97.8 |
| SAGE (Ours) | 2048 | 95.6 | 99.2 | **99.7** | 95.4 | 97.4 | 97.9 | 92.6 | 96.9 | **97.7** | 91.2 | 96.6 | 97.8 |
| | 4096 | **97.7** | **99.8** | **100** | **95.6** | **97.7** | **98.3** | 93.7 | **97.3** | **97.8** | 94.4 | 98.2 | 99.0 |
| | 8448 | **98.9** | **99.7** | **100** | **95.8** | **97.8** | **98.4** | **94.5** | **97.4** | **97.8** | **96.0** | **98.9** | **99.4** |

Table 3: Comparison to SoTA methods on more challenging datasets. Values marked with [‡] were reproduced in this work when they were not reported in the original publications.

| Method | Dim | AmsterTime | | | Tokyo24/7 | | | Pitts250k-test | | | Eynsham | | |
|---|---|---|---|---|---|---|---|---|---|---|---|---|---|
| | | R@1 | R@5 | R@10 | R@1 | R@5 | R@10 | R@1 | R@5 | R@10 | R@1 | R@5 | R@10 |
| SALAD-CM ECCV'2024 | 8448 | 57.8 | 77.5 | 81.3 | **96.8** | 97.5 | 97.8 | 95.2 | 98.8 | 99.3 | 91.9 | 95.3 | 96.1 |
| SuperVLAD NIPS'2024 | 3072 | 63.9 | 83.9 | 87.3 | 95.6 | 97.8 | 98.1 | 97.3 | **99.4** | **99.7** | 92.1 | 95.6 | 96.4 |
| EMVP NIPS'2024 | 8448 | 65.6[‡] | 86.0[‡] | 90.5[‡] | **96.8**[‡] | 98.1[‡] | **98.7**[‡] | 96.5 | **99.1** | **99.5** | 91.9[‡] | 95.7[‡] | 96.6[‡] |
| FoL AAAI'2025 | 8448 | 64.6 | 84.3 | 88.2 | 96.2 | **98.7** | **98.7** | 96.5 | **99.1** | **99.5** | 91.7 | 95.3 | 96.2 |
| SAGE (Ours) | 2048 | 66.2 | 78.6 | 85.0 | 95.6 | 96.5 | 98.1 | 97.7 | **99.1** | 99.3 | 92.7 | 95.8 | 96.5 |
| | 4096 | **76.0** | **88.0** | **92.3** | 96.5 | **99.1** | **99.4** | **98.2** | **99.4** | **99.5** | **92.9** | **96.0** | **96.8** |
| | 8448 | **83.5** | **93.3** | **95.4** | **97.5** | **99.1** | **99.4** | **98.4** | **99.4** | **99.7** | **93.1** | **96.2** | **97.0** |

## 4.2 IMPLEMENTATION DETAILS

SAGE is built upon the EMVP framework (Qiu et al., 2024), which we reproduced from its publication for a fair comparison as the official code is unavailable. We fine-tune two Vision Transformer backbones, ViT-B and ViT-L, which we denote as SAGE-B and SAGE-L, respectively. The Feature Compression ($\mathcal{F}_C$) and Feature Probing ($\mathcal{F}_P$) branches are implemented as two-layer MLPs, reducing the feature dimensions to $D = 128$ and $K = 64$, respectively. The InteractHead module is im-

Table 1: Summary of the evaluation datasets.

| Dataset | Description | Number | |
|---|---|---|---|
| | | Database | Queries |
| Pitts30k-test | urban, panorama | 10,000 | 6,816 |
| MSLS-val | urban, suburban | 18,871 | 740 |
| Nordland | natural, seasonal | 27,592 | 27,592 |
| SPED | various scenes | 607 | 607 |
| Tokyo24/7 | urban, time-varying | 75,984 | 315 |
| AmsterTime | urban, time-related | 1,231 | 1,231 |
| Pitts250k-test | urban, panorama | 83,952 | 8,280 |
| Eynsham | rural, historical | 23,935 | 23,935 |

plemented as a two-layer Transformer encoder with a model dimension of 768, 16 attention heads, and a feed forward network with dimension 1024. During training we freeze the backbone and adopt DPN for PEFT, which adaptively preserves task-specific information while greatly reducing the number of trainable parameters. Each training mini-batch is constructed with equal contribu-

tions from MSLS (all non-panoramic images from the training set) and GSV-Cities (0.56M images from 67K places). For every batch we build a new sparse geo-visual graph, and we sample $P = 15$ sequences from the same city and recompute cliques dynamically during training. The thresholds for this process are set to $\tau_1 = 25$ m and $\tau_2 = -2.88 \times 10^3$. Input images are resized to $224 \times 224$ during training and $322 \times 322$ during inference. All experiments are implemented in PyTorch and run efficiently on a single NVIDIA A100 GPU. We fine-tune models for 10 epochs and select the checkpoint with the highest Recall@1 on Pitts30k-test for evaluation on the other benchmarks.

## 4.3 COMPARISONS WITH SOTA METHODS

In this section we provide a comprehensive comparison between our proposed SAGE and a range of state-of-the-art VPR methods. The comparison includes: NetVLAD (Arandjelovic et al., 2016), SFRS (Ge et al., 2020), CosPlace (Berton et al., 2022a), MixVPR (Ali-Bey et al., 2023b), Eigen-Places (Berton et al., 2023), CricaVPR (Lu et al., 2024b), SALAD (Izquierdo & Civera, 2024), BOQ (Ali-bey et al., 2024), SALAD-CM (Izquierdo & Civera, 2025), SuperVLAD (Lu et al., 2024d), EMVP (Qiu et al., 2024), and the two re-ranking (two-stage) pipelines SelaVPR (Lu et al., 2024c) and FoL (Wang et al., 2025). Re-ranking pipelines add a computationally intensive local feature matching stage. EMVP and FoL are the current SOTA for single-stage global retrieval and two-stage VPR pipelines, respectively; implementation details for compared methods are given in App. A.4.

As shown in Tab. 2 and Tab. 3, SAGE consistently outperforms previous methods across all benchmarks and evaluation metrics. In a higher-dimensional configuration (8448-d), SAGE reaches 94.5% R@1 on MSLS-val and achieves 100% R@10 on SPED, while improving R@1 by 4.3 percentage points over the previous best single-stage method (EMVP). SAGE also maintains leading performance on challenging datasets: for example, it attains 96.0% R@1 on Nordland and 83.5% R@1 on AmsterTime, representing substantial gains relative to FoL and EMVP. We also evaluate compact configurations obtained via PCA dimensionality reduction. Even under tighter budgets (2048-d and 4096-d), SAGE remains highly competitive and often matches or even surpasses recent strong baselines. For instance, the 4096-d SAGE achieves 95.6% R@1 on Pitts30k-test and 97.7% R@1 on SPED while preserving 100% R@10, demonstrating that our proposed feature amplification and epoch-wise online geo-visual sampling produce highly discriminative global descriptors. A detailed analysis of SAGE performance across varying descriptor dimensions is presented in App. A.2.

We apply t-SNE to embed spatial features from four methods into a 2D space for visual comparison. Features are extracted from 600 images at 50 locations. Fig. 3 displays the 2D data projections and the corresponding Average Intra-class Distance (AID), according to Equ. 8. SAGE-B has the smallest AID indicating the tightest within location clustering. More results is presented in Fig. 6.

Tab. 4 details parameters for various VPR methods. By employing DPN for PEFT, SAGE avoids the heavy adapter modules typical of methods like SelaVPR and CricaVPR, thus achieving a markedly lower total parameters. More strikingly, since SAGE keeps the backbone frozen and exclusively fine-tunes its lightweight DPN, SoftP, and InteractHead modules, its **trainable** parameters is significantly smaller than that of approaches that fine-tune portions of the Transformer encoder (e.g., SALAD, SALAD-CM, and SuperVLAD). This high parameter efficiency is attained without even compromising its SOTA performance.

Table 4: Comparison of parameters (M) for various VPR methods using the DINOv2-B backbone. The value in parentheses is the number of parameters in the optional cross-image encoder.

| Method | Total ↓ | Trainable ↓ | Adapter |
|---|---|---|---|
| SALAD CVPR' 2024 | **88.0** | 29.8 | ✗ |
| SelaVPR ICLR' 2024 | 102.8 | 16.2 | ✔ 14.2 |
| CricaVPR CVPR' 2024 | 95.7 (+11.0) | 9.15 (+11.0) | ✔ 9.2 |
| SALAD-CM ECCV' 2024 | **88.0** | 29.8 | ✗ |
| SuperVLAD NIPS' 2024 | 86.6 (+11.0) | 28.4 (+11.0) | ✗ |
| EMVP NIPS' 2024 | 88.5 | **1.96** | ✗ |
| SAGE (Ours) | 88.5 (+7.88) | **1.96** (+7.88) | ✗ |

Fig. 4 shows qualitative results for SAGE-B and several SOTA methods in representative challenging scenarios. SAGE consistently retrieves the correct database images, while other methods often fail to capture the most discriminative cues and produce incorrect matches. Additional qualitative visualizations are provided in App. A.1, as illustrated in Fig. 7. To illustrate the comparison, we show importance heatmaps produced by SoftP, SALAD, and CFP in Fig. 5. While all three highlight prominent static landmarks, but SoftP even more effectively concentrates on subtle, fine-grained, highly discriminative regions. Additional SoftP heatmaps are provided in App. A.1, shown in Fig. 8.

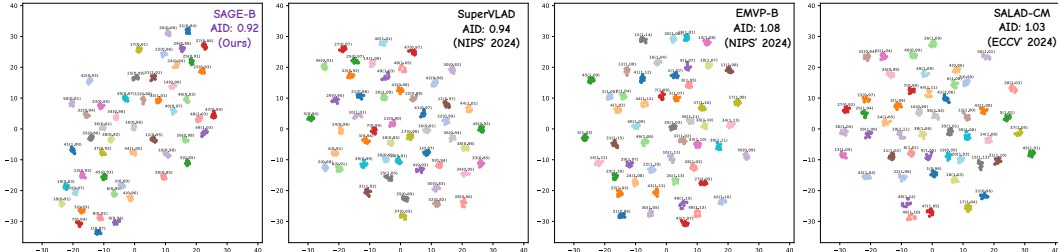

Figure 3: Visualization of spatial feature clustering using t-SNE for four methods and comparison of Average Intra-class Distance (AID). Numbers next to each class indicate intra-class distance (ID).

Table 5: Ablation of SAGE components. All experiments use ViT-B; results reproduced in this work are marked ‡. OGC denotes Online Graph Creation and GWS denotes Greedy Weighted Sampling.

| Method | Components | | | SPED | | | Pitts30k-test | | | MSLS-val | | | Nordland | | |
|---|---|---|---|---|---|---|---|---|---|---|---|---|---|---|---|
| | Aggregation | OGC | GWS | R@1 | R@5 | R@10 | R@1 | R@5 | R@10 | R@1 | R@5 | R@10 | R@1 | R@5 | R@10 |
| EMVP-B | CFP | | | 91.8 | 96.5 | 97.4 | 93.1‡ | 96.8‡ | 97.6‡ | 93.2 | 96.9 | 97.2 | 80.8‡ | 90.4‡ | 93.5‡ |
| SAGE-B | SoftP | ✓ | | 96.8 | 98.2 | 98.7 | 94.6 | 97.2 | 97.9 | 93.6 | 96.8 | 97.1 | 95.2 | 98.4 | 98.7 |
| | SoftP | | ✓ | 96.5 | 97.8 | 98.3 | 93.8 | 96.5 | 97.2 | 92.5 | 96.6 | 96.9 | 94.2 | 97.4 | 97.9 |
| | CFP | ✓ | ✓ | 97.5 | 98.4 | 98.9 | 94.9 | 97.3 | 98.0 | 93.9 | 97.1 | 97.4 | 95.4 | 98.5 | 98.8 |
| | SoftP | ✓ | ✓ | **98.0** | **98.7** | **99.2** | **95.4** | **97.6** | **98.3** | **94.3** | **97.2** | **97.6** | **95.8** | **98.7** | **99.2** |

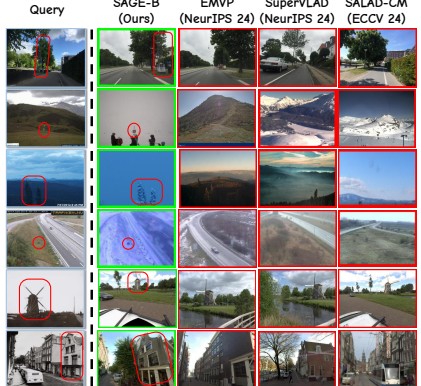

Figure 4: Qualitative results. SAGE consistently retrieves correct database images under severe challenges.

Figure 5: Visual comparison of importance heatmaps. SoftP shows a stronger focus on fine grained regions with high discriminative value than other methods overall.

## 4.4 ABLATION STUDY

We ablate SAGE's components using a ViT-B backbone (8448-D) and EMVP-B as the baseline (Tab. 5). On the challenging Nordland dataset, marked by severe seasonal variations, the baseline only achieves 80.8% R@1. Integrating our SoftP and OGC modules yields substantial gains, boosting R@1 to 93.6 on MSLS-val and 95.2% on Nordland. These gains suggest that SoftP enhances discriminative local feature responses, while OGC reconstructs a geo–visual graph each training epoch to expose the model to evolving hard examples aligned with the current embedding space.

Adding GWS to a SoftP configuration produces modest and unstable gains, suggesting sampling alone cannot exploit graph dynamics without Online Graph Creation. Enabling OGC and GWS while retaining CFP yields a notable improvement, for example Pitts30k-test R@1 of 94.9%, but still falls short of the configuration that includes SoftP. The full SAGE configuration achieves the best results, with R@1 of 95.4% on Pitts30k-test and R@1 of 98.0% on SPED, surpassing the baseline and intermediate variants. The greedy weighted clique expansion complements SoftP and OGC by focusing training on the most informative clusters and enhancing descriptor discriminability.

To evaluate the computational cost of our dynamic sampling strategy, we conducted an ablation study comparing the runtime and performance of online versus offline graph creation. As shown in Table 6, our online approach incurs a modest 17.7% increase in per-epoch training time. However, this modest overhead is a worthwhile investment, as it translates directly to su-

Table 6: Comparison of Online and Offline Graph Creation Strategies. Runtimes are reported per epoch for the online method. For the offline strategy, mining is a one-time cost.

| Strategy | Method | Mining (min) | Train (min) | SPED | | | MSLS-val | | |
|----------|--------|--------------|-------------|------|------|-------|----------|------|-------|
| | | | | R@1 | R@5 | R@10 | R@1 | R@5 | R@10 |
| Online | Cliquemining | 4.3 | 25.1 | 90.0 | 95.4 | 96.2 | 94.3 | 96.9 | 97.6 |
| | SAGE (w/o GWS) | 6.1 | 28.4 | 96.8 | 98.2 | 98.7 | 93.6 | 96.8 | 97.1 |
| | SAGE | 6.2 | 28.4 | **98.9** | **99.7** | **100** | **94.5** | **97.4** | **97.8** |
| Offline | Cliquemining | 21.6 | 25.1 | 89.5 | 94.9 | 96.1 | 94.2 | 97.2 | 97.4 |
| | SAGE (w/o GWS) | 30.7 | 28.4 | 96.7 | 98.2 | 98.9 | 93.5 | 96.6 | 97.1 |
| | SAGE | 30.9 | 28.4 | **98.5** | **99.3** | **99.5** | **94.2** | **97.3** | **97.7** |

perior accuracy, with the online SAGE model achieving higher recall on both SPED and MSLS-val.

This result validates our central argument: dynamically adapting the sampling to the model's evolving state is crucial for breaking the performance bottleneck of static mining strategies. Furthermore, the experiment highlights the efficiency of the GWS module itself, which adds negligible overhead while delivering a substantial accuracy boost. Crucially, this computational overhead is confined to the training phase. The inference process is un-

Table 7: Convergence analysis on MSLS-val. SAGE's dynamic sampling leads to superior performance in early training epochs.

| Epoch | SAGE (w/ CM) | | | SAGE (Ours) | | |
|-------|------|------|------|------|------|------|
| | R@1 | R@5 | R@10 | R@1 | R@5 | R@10 |
| 2 | 92.3 | 96.1 | 96.6 | 92.5 | 96.5 | 97.1 |
| 4 | 92.7 | 96.6 | 97.0 | **93.4** | **96.9** | **97.4** |

affected, ensuring SAGE remains as efficient as comparable single-stage methods at deployment.

To demonstrate SAGE's learning efficiency, we tracked its early-stage training performance on MSLS-val against a baseline using the Cliquemining (CM) strategy. As shown in Table 7, our model establishes a clear advantage by the fourth epoch (93.4% vs. 92.7% at R@1). The widening performance gap confirms that our dynamic sampling strategy fosters more efficient learning, leading to superior performance within the same number of epochs. In

Table 8: Ablation on InteractHead module. We vary the model dimension ($d_{\mathrm{model}}$) and feed-forward dimension ($d_{\mathrm{ff}}$).

| $(d_{\mathrm{model}}, d_{\mathrm{ff}})$ | SPED | | | Pitts30k | | |
|-------|------|------|------|------|------|------|
| | R@1 | R@5 | R@10 | R@1 | R@5 | R@10 |
| (512, 1024) | 98.8 | 99.5 | 99.7 | 95.5 | 97.6 | 98.2 |
| (768, 1536) | 98.6 | 99.3 | 99.7 | **96.0** | **97.9** | **98.4** |
| (768, 1024) | **98.9** | **99.7** | **100** | 95.8 | 97.8 | **98.4** |

Tab. 8, we ablate the InteractHead's internal dimensions. Setting $d_{model} = 768$ and $d_{ff} = 1024$ yields the best performance, notably achieving 98.9% R@1 on SPED. Expanding $d_{ff}$ to 1536 marginally improves Pitts30k but degrades SPED. Thus, we adopt the (768, 1024) configuration, balancing capacity and generalization. Extended comparisons with recent SOTA methods (Table 11) and additional evaluation results without the InteractHead (Table 14) are detailed in App. A.2.

## 5 CONCLUSION

In this paper, we presented SAGE, a unified framework that redefines Visual Place Recognition training by shifting from static sampling strategies to a dynamic, slow thinking paradigm. By synergizing the lightweight Soft Probing module with InteractHead, SAGE effectively amplifies fine-grained discriminative cues and models cross-image correlations, ensuring robust feature representation even under drastic appearance variations. Crucially, our novel Online Graph Creation and Greedy Weighted Sampling mechanisms ensure that the mining process continuously synchronizes with the evolving embedding space, allowing the model to relentlessly focus on the most informative geo-visual neighborhoods. Extensive evaluations across eight diverse benchmarks demonstrate that SAGE establishes a new SOTA, delivering exceptional retrieval accuracy while maintaining remarkable parameter efficiency through a frozen DINOv2 backbone. This provides a scalable and efficient foundation for future large-scale visual geo-localization systems.

## 6 ACKNOWLEDGMENTS

This work was supported by the Beijing Natural Science Foundation (No. JQ23014) and the National Natural Science Foundation of China (No. 62271074); and in part by the BUPT Kunpeng&Ascend Center of Cultivation, the Taishan Scholars Program (No. TSQN202507241), the Key R&D Program of Shandong Province, China (No. 2025KJHZ013), the Shandong Provincial University Youth Innovation and Technology Support Program (No. 2022KJ291), the Shandong Provincial Natural Science Foundation for Young Scholars Program (No. ZR2025QC1627), and the Qilu University of Technology (Shandong Academy of Sciences) Youth Outstanding Talent Program (No. 2024QZJH02).

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

# A APPENDIX

## A.1 VISUALIZATION OF RESULTS

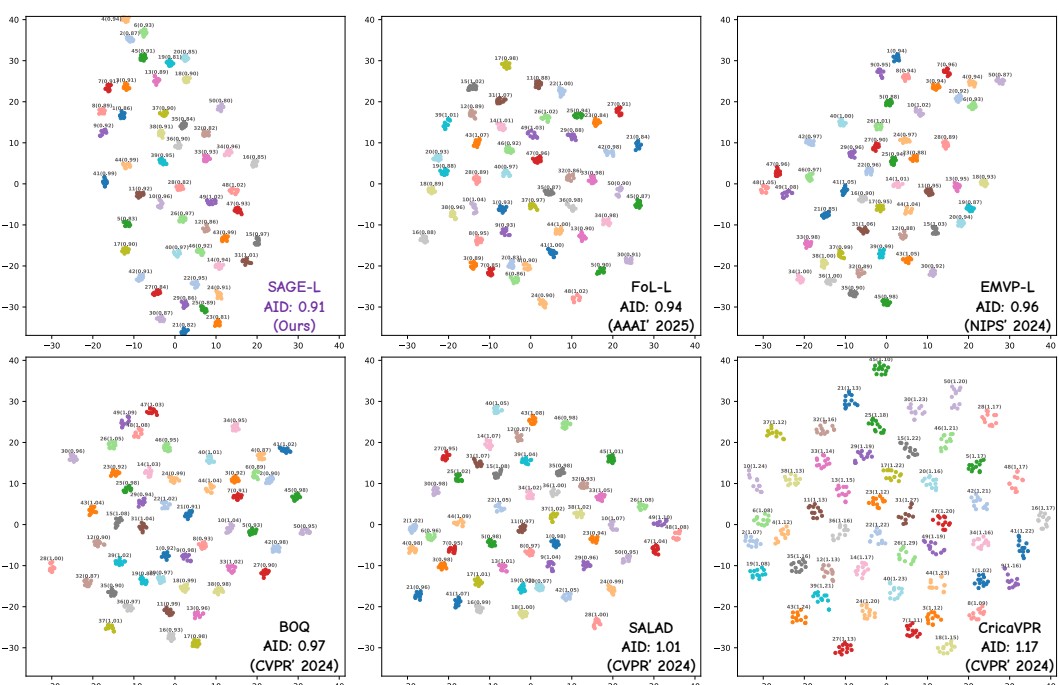

Figure 6: t-SNE visualization of feature clusters produced by SAGE-L and five other leading VPR methods. The features are extracted from 600 images across 50 distinct locations from the GSV-Cities dataset. Clustering compactness is quantitatively evaluated using the Average Intra-class Distance (AID), defined as the mean Euclidean distance of features to their corresponding class centroid. A lower AID signifies a more discriminative and compact feature representation for images of the same location. Notably, SAGE-L achieves the lowest AID (0.91), demonstrating its superior ability to group features from the same place.

To visually assess the quality of feature clustering, we first created a dedicated test set by selecting 600 images from 50 distinct locations (12 images per location) within the GSV-Cities dataset, ensuring coverage of diverse scenes and conditions. We then employed t-SNE to project the high-dimensional features generated by each method into a 2D space for visualization.

For a quantitative analysis of cluster compactness, we calculated the Average Intra-class Distance (AID). This metric is derived from the Intra-class Distance (ID), which measures the mean Euclidean distance of features within a class to their shared centroid $\mu_i$, as defined by:

$$\text{ID}_i = \frac{1}{|C_i|} \sum_{x \in C_i} \|x - \mu_i\|_2, \qquad \text{AID} = \frac{1}{N} \sum_{i=1}^{N} \text{ID}_i. \tag{8}$$

where $C_i$ is the set of feature vectors for location $i$. A lower AID value signifies a more compact and discriminative feature representation. As illustrated in Fig. 6, SAGE-L achieves the lowest AID, which quantitatively confirms its superior ability to generate robust features that are tightly clustered for the same location, effectively handling intra-class variations.

To highlight the practical robustness of our method, Fig. 7 presents a qualitative hcomparison between SAGE-B and seven leading VPR methods. The visualization is structured to systematically evaluate retrieval performance across six of the most common and difficult VPR challenges: severe viewpoint shifts, adverse weather, drastic lighting changes, long-term temporal differences, structural alterations, and dynamic occlusions. In these demanding scenarios, most state-of-the-art methods falter, failing to identify the correct place and retrieving visually plausible but incorrect

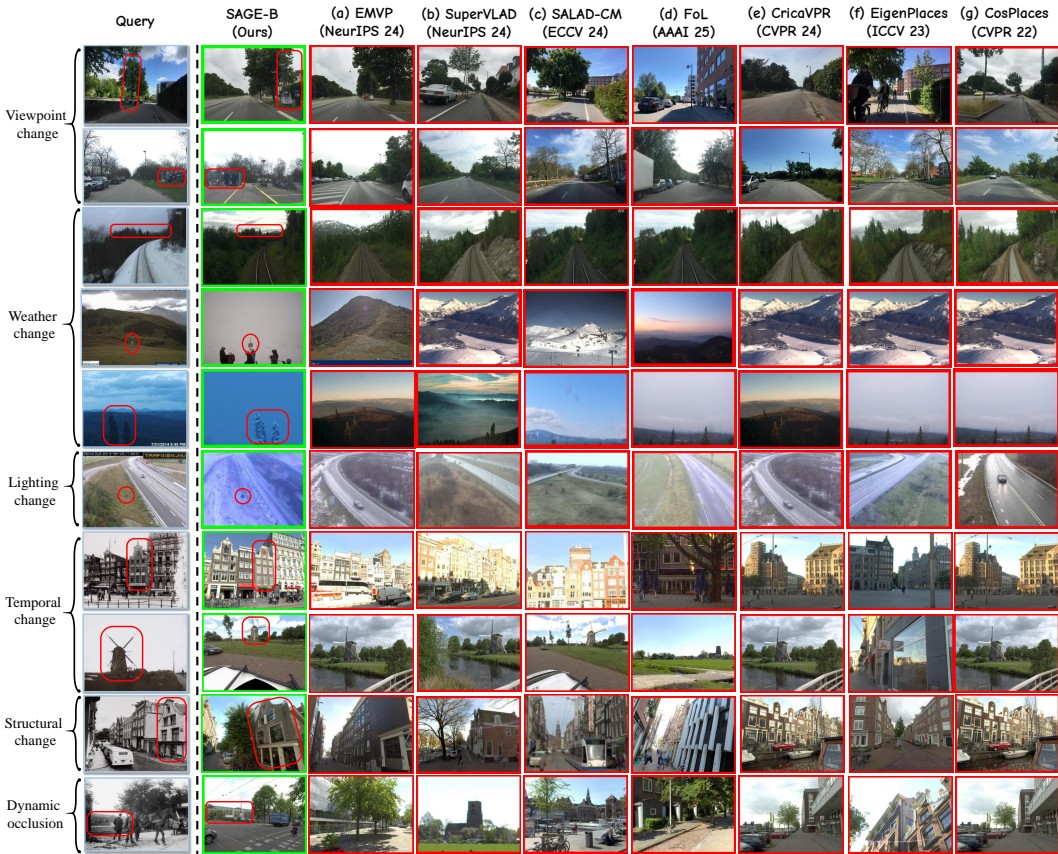

Figure 7: Qualitative comparison of SAGE-B against leading VPR methods under diverse and challenging conditions. Rows correspond to challenge category, from top to bottom: viewpoint change, weather change, lighting change, temporal change, structural change, and dynamic occlusion. Correct top-1 retrievals are indicated by a green bounding box, while incorrect matches are marked in red. The results visually confirm SAGE's consistent and superior robustness across all scenarios.

matches (highlighted in red). In stark contrast, SAGE-B consistently retrieves the correct database image in every case (green boxes). This demonstrates the superior resilience of our approach, which stems from its ability to learn and focus on stable, truly discriminative features while effectively mitigating the impact of significant appearance variations.

The heatmap visualizations of the SoftP module, presented in Figure 8, elucidate its underlying mechanism. A clear pattern is evident across diverse scenes where the model learns to automatically suppress features from non-informative or transient sources, including the sky, road surfaces, and dynamic objects such as vehicles and pedestrians. Crucially, SoftP moves beyond concentrating on large static structures to prioritize fine-grained, stable details that offer reliable discriminative cues, for instance specific architectural features, window frames, and unique textures.

## A.2 ADDITIONAL RESULTS

To further demonstrate the scalability and robustness of our SAGE framework, we present an performance analysis across a range of descriptor dimensions. This analysis spans both standard and more challenging VPR benchmarks, highlighting the consistent effectiveness of SAGE.

Tab. 9 details the performance of SAGE on four widely used VPR benchmarks (SPED, Pitts30k-test, MSLS-val, and Nordland) with descriptor dimensions varying from 128 to 8448. The results reveal a clear and consistent trend: retrieval accuracy, measured by Recall@N, systematically improves as the descriptor dimension increases. Notably, even at an intermediate dimension of 4096, SAGE achieves remarkable performance, including a perfect 100% R@10 on SPED. The results at 8448-D,

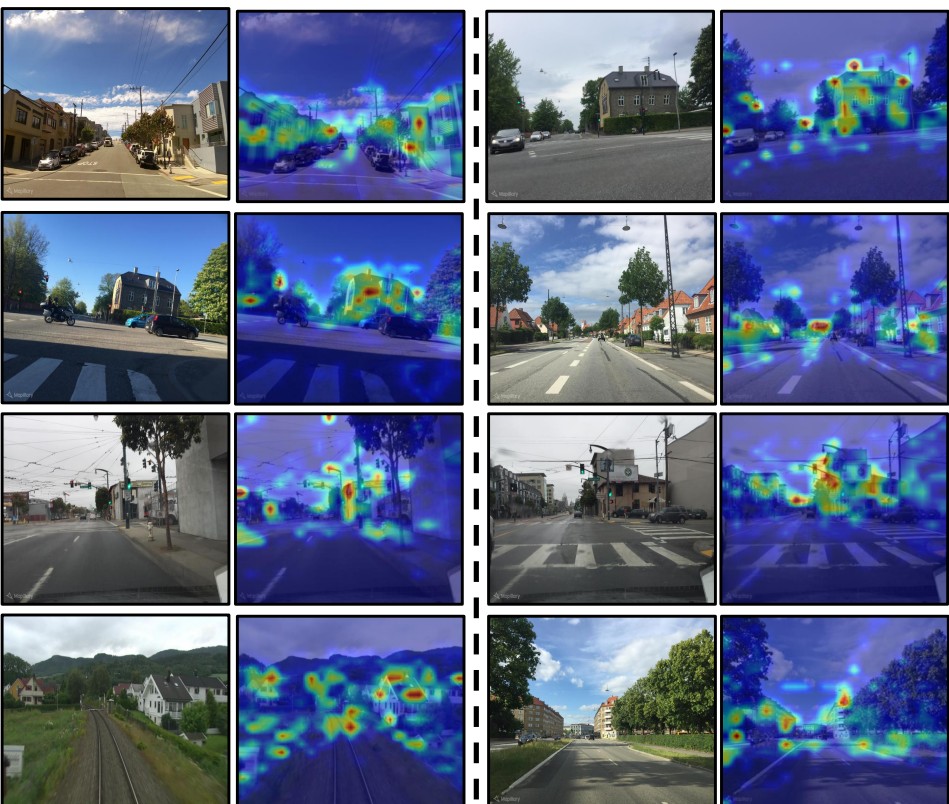

Figure 8: Visualization of SoftP's learned feature importance. The visualizations demonstrate that SoftP automatically learns to focus on stable, fine-grained landmarks (e.g., building facades, structural details) while effectively ignoring non-discriminative regions (sky, road) and transient objects (vehicles, pedestrians).

Table 9: Performance analysis of SAGE across varying descriptor dimensions. Results on VPR benchmarks show a consistent improvement in retrieval accuracy as the dimension increases. The best and second best results for each dataset are shown in **red bold** and **blue bold**, respectively.

| Method | Dim | SPED | | | Pitts30k-test | | | MSLS-val | | | Nordland | | |
|---|---|---|---|---|---|---|---|---|---|---|---|---|---|
| | | R@1 | R@5 | R@10 | R@1 | R@5 | R@10 | R@1 | R@5 | R@10 | R@1 | R@5 | R@10 |
| SAGE (Ours) | 128 | 84.7 | 92.8 | 95.7 | 88.9 | 94.5 | 95.7 | 81.8 | 91.5 | 93.0 | 56.1 | 72.2 | 78.1 |
| | 256 | 91.6 | 96.7 | 97.5 | 93.0 | 95.9 | 96.7 | 87.2 | 94.6 | 95.7 | 70.5 | 83.6 | 87.8 |
| | 512 | 94.2 | 98.2 | 99.0 | 94.4 | 96.5 | 97.2 | 91.4 | 95.7 | 96.8 | 80.1 | 90.0 | 92.9 |
| | 1024 | 95.4 | 99.3 | 99.7 | 95.1 | 97.1 | 97.6 | 92.4 | 96.5 | 96.9 | 87.0 | 94.3 | 96.2 |
| | 2048 | 95.6 | 99.2 | 99.7 | 95.4 | 97.4 | 97.9 | 92.6 | 96.9 | **97.7** | 91.2 | 96.6 | 97.8 |
| | 3072 | 96.9 | **99.7** | **99.8** | **95.6** | **97.7** | 98.2 | 92.4 | **97.3** | **97.7** | 93.8 | 97.8 | 98.8 |
| | 4096 | **97.7** | **99.8** | **100** | **95.6** | **97.7** | **98.3** | **93.7** | **97.3** | **97.8** | **94.4** | **98.2** | **99.0** |
| | 8448 | **98.9** | **99.7** | **100** | **95.8** | **97.8** | **98.4** | **94.5** | **97.4** | **97.8** | **96.0** | **98.9** | **99.4** |

such as 98.9% R@1 on SPED and 96.0% R@1 on Nordland, underscore the framework's ability to leverage higher-dimensional feature spaces for enhanced discriminability.

Building on this, Tab. 10 extends the evaluation to more demanding datasets characterized by severe domain shifts, including AmsterTime (historical vs. modern), Tokyo24/7 (extreme viewpoint and time-of-day changes), Pitts250k-test (large-scale urban scenes), and Eynsham (rural route). The performance trend remains robust, with accuracy scaling gracefully with descriptor dimensionality. The improvement is particularly pronounced on AmsterTime, where the R@1 score surges from 23.4% at 128-D to 83.5% at 8448-D. This demonstrates SAGE's exceptional capability to handle extreme appearance variations, validating the effectiveness of our proposed dynamic geo-visual graph exploration and feature enhancement strategies.

Table 10: Performance of SAGE with varying descriptor dimensions on more challenging datasets.

| Method | Dim | AmsterTime | | | Tokyo24/7 | | | Pitts250k-test | | | Eynsham | | |
|---|---|---|---|---|---|---|---|---|---|---|---|---|---|
| | | R@1 | R@5 | R@10 | R@1 | R@5 | R@10 | R@1 | R@5 | R@10 | R@1 | R@5 | R@10 |
| SAGE (Ours) | 128 | 23.4 | 36.5 | 44.4 | 42.9 | 60.6 | 66.4 | 89.0 | 95.1 | 96.4 | 89.3 | 93.2 | 94.3 |
| | 256 | 36.0 | 51.0 | 58.7 | 66.0 | 78.7 | 82.5 | 94.7 | 97.7 | 98.2 | 91.0 | 94.4 | 95.1 |
| | 512 | 44.9 | 60.8 | 68.1 | 80.0 | 89.2 | 92.1 | 96.4 | 98.3 | 98.7 | 91.9 | 95.0 | 95.7 |
| | 1024 | 55.6 | 70.0 | 76.9 | 89.5 | 94.6 | 95.9 | 97.3 | 98.9 | 99.2 | 92.4 | 95.5 | 96.2 |
| | 2048 | 66.2 | 78.6 | 85.0 | 95.6 | 96.5 | 98.1 | 97.7 | 99.1 | 99.3 | 92.7 | 95.8 | 96.5 |
| | 3072 | 73.6 | 85.8 | 89.9 | 94.9 | **98.4** | **98.7** | 98.1 | **99.3** | 99.5 | **92.9** | **96.0** | 96.7 |
| | 4096 | **76.0** | **88.0** | **92.3** | **96.5** | **99.1** | **99.4** | **98.2** | **99.4** | 99.5 | **92.9** | **96.0** | **96.8** |
| | 8448 | 83.5 | 93.3 | 95.4 | 97.5 | 99.1 | 99.4 | 98.4 | 99.4 | 99.7 | 93.1 | 96.2 | 97.0 |

To further validate the effectiveness of SAGE, we present an extended comparison against several recently published SOTA methods. These include VLAD-BuFF (Khaliq et al., 2024), EDTFormer (Jin et al., 2025b), MegaLoc (Berton & Masone, 2025), ImAge (Lu et al., 2025b), SelaVPR (Lu et al., 2024c), EffoVPR (Tzachor et al., 2025), FoL (Wang et al., 2025), and SelaVPR++ (Lu et al., 2025a). The results, shown in Table 11, demonstrate that SAGE consistently achieves highly competitive or superior performance against these strong baselines. Although some methods like EffoVPR and SelaVPR++ obtain slightly higher results on specific metrics (e.g., R@1 on Tokyo24/7), they are two-stage reranking methods, which unavoidably introduce additional computational overhead.

Table 11: Extended comparison with recent SOTA on five challenging benchmarks. R1, R5, and R10 represent R@1, R@5, and R@10, respectively. [†]Denotes two-stage methods (reranking results only). -B and -L denote ViT-B and ViT-L, respectively. Except for EffoVPR, other two-stage methods use DINOv2 without registers. Furthermore, while EffoVPR is tested at a resolution of $504 \times 504$, all other methods (including FoL) are evaluated under a consistent $322 \times 322$ setting.

| Method | SPED | | | MSLS-val | | | Nordland | | | Tokyo24/7 | | | Pitts250k-test | | |
|---|---|---|---|---|---|---|---|---|---|---|---|---|---|---|---|
| | R1 | R5 | R10 | R1 | R5 | R10 | R1 | R5 | R10 | R1 | R5 | R10 | R1 | R5 | R10 |
| VLAD-BuFF (-B) ECCV' 24 | 91.4 | 95.9 | 96.9 | 91.8 | 96.0 | 96.2 | 85.1 | 93.8 | 96.0 | 96.2 | 98.7 | 99.4 | 95.5 | 98.5 | 99.2 |
| EDTFormer (-B) TCSVT' 25 | 92.4 | 95.9 | 96.9 | 92.0 | 96.6 | 97.2 | 88.3 | 95.3 | 97.0 | 97.1 | 98.1 | 98.4 | 95.9 | 98.8 | 99.3 |
| FoL-global (-B) AAAI' 25 | 89.5 | 95.2 | 96.4 | 91.1 | 95.7 | 96.4 | 72.7 | 85.5 | 89.6 | 94.6 | 96.5 | 96.8 | 94.8 | 98.6 | 99.2 |
| FoL-global (-L) AAAI' 25 | 91.8 | 96.2 | 97.9 | 92.4 | 96.9 | 97.2 | 83.8 | 92.6 | 95.1 | 96.5 | 98.1 | 98.4 | 96.2 | 99.0 | 99.4 |
| MegaLoc (-B) CVPRW'25 | 90.3 | 95.4 | 95.9 | 93.5 | 97.0 | 97.8 | 94.2 | 97.9 | 98.6 | 96.5 | 98.1 | **99.7** | 96.4 | 98.9 | 99.3 |
| ImAge (-B) NIPS' 25 | 91.6 | 95.6 | 96.7 | 93.0 | 97.0 | 97.2 | 93.2 | 97.6 | 98.6 | 96.2 | 98.1 | 98.4 | 96.5 | 99.1 | 99.5 |
| SelaVPR (-L) ICLR' 24 [†] | 88.6 | 95.1 | 97.2 | 90.8 | 96.4 | 97.2 | 87.3 | 93.8 | 95.6 | 94.0 | 96.8 | 97.5 | 95.7 | 98.8 | 99.2 |
| EffoVPR (-L) ICLR' 25 [†] | 93.1 | 97.9 | 98.4 | 92.8 | 97.2 | 97.4 | 95.0 | - | - | **98.7** | 98.7 | 98.7 | 97.0 | - | - |
| FoL (-B) AAAI' 25 [†] | 90.6 | 94.4 | 95.6 | 91.5 | 96.2 | 96.8 | 85.4 | 92.7 | 94.8 | 97.5 | 98.1 | 98.4 | 96.1 | 99.1 | 99.5 |
| FoL (-L) AAAI' 25 [†] | 92.4 | 96.9 | 97.5 | 90.1 | 95.7 | 96.9 | 87.9 | 94.8 | 96.6 | 97.1 | 97.8 | 98.7 | 96.8 | 99.0 | 99.4 |
| SelaVPR++ (-B) TPAMI' 25 [†] | 90.6 | 95.2 | 96.2 | 94.3 | 97.2 | 97.3 | 94.9 | 97.8 | 98.4 | 97.5 | 99.0 | 99.0 | 95.9 | 98.6 | 99.2 |
| SelaVPR++ (-L) TPAMI' 25 [†] | 92.6 | 96.5 | 97.0 | **94.5** | **98.0** | **98.2** | **97.2** | **99.0** | **99.4** | 98.1 | 98.7 | 99.4 | 96.6 | 99.0 | 99.3 |
| SAGE (Ours) | **98.9** | **99.7** | **100** | **94.5** | 97.4 | 97.8 | 96.0 | 98.9 | **99.4** | 97.5 | **99.1** | 99.4 | **98.4** | **99.4** | **99.7** |

To further assess generalization, we conducted experiments on the SF-small benchmark. Derived from SF-XL (Berton et al., 2022a), this dataset is particularly well-suited for evaluating model generalization as its queries are drawn from distinct, non-adjacent geographic locations. The results are presented in Table 12. SAGE achieves 89.3% at R@1, significantly outperforming all compared state-of-the-art methods. Our method utilizes two hyperparameters during Online Graph Creation: the geographic distance threshold $\tau_1$ and the affinity score threshold $\tau_2$. We conducted a sensitivity analysis on these parameters using the MSLS-val dataset, as presented in Table 13. The table also shows that the model's performance remains

Table 12: Performance comparison on the SF-small benchmark.

| Method | R@1 | R@5 | R@10 |
|---|---|---|---|
| CricaVPR CVPR' 2024 | 84.5 | 88.7 | 89.2 |
| SALAD CVPR' 2024 | 85.7 | 88.2 | 89.7 |
| SuperVLAD NIPS' 2024 | 85.8 | 89.1 | 89.5 |
| SALAD-CM ECCV' 2024 | 84.0 | 88.0 | 89.8 |
| EDTformer TCSVT' 2025 | 87.9 | 89.8 | 90.6 |
| EMVP NIPS' 2024 | 88.2 | 90.6 | 91.1 |
| SAGE (Ours) | **89.3** | **91.5** | **91.9** |

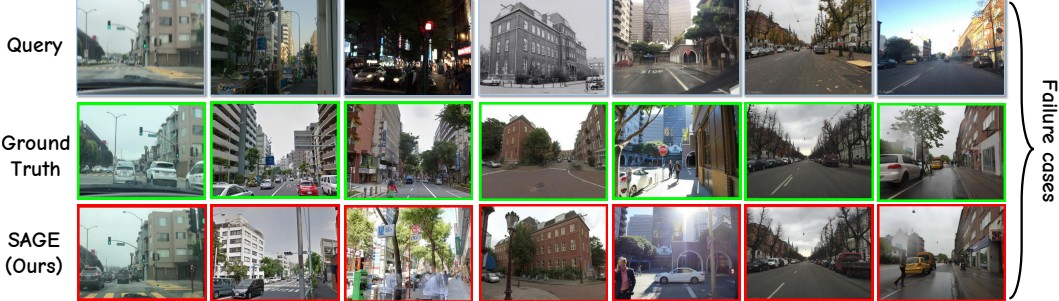

Figure 9: Failure cases of SAGE, where the top-1 retrieval (red box) fails to match the ground truth.

stable around these optimal values, demonstrating that our method is not overly sensitive to these hyperparameters and exhibits good robustness.

We present the evaluation results of our models (SAGE-B and SAGE-L) without the InteractHead module. This experiment is conducted uniformly at a resolution of $322 \times 322$. As shown in Table 14, even without modeling cross-image correlations within a batch, our method continues to demonstrate highly competitive retrieval performance across ten diverse VPR datasets. Specifically, the ViT-L variant

Table 13: Sensitivity analysis of $\tau_1$ and $\tau_2$.

| $\tau_1$ | MSLS-val | | | $\tau_2$ | MSLS-val | | |
|---|---|---|---|---|---|---|---|
| | R@1 | R@5 | R@10 | | R@1 | R@5 | R@10 |
| 20 | 93.8 | 97.3 | 97.7 | $-2.75 \times 10^3$ | 94.1 | 97.6 | 97.8 |
| 25 | **94.5** | 97.4 | 97.8 | $-2.88 \times 10^3$ | **94.5** | 97.4 | 97.8 |
| 30 | 93.7 | 97.0 | 97.4 | $-3.00 \times 10^3$ | 93.9 | 97.3 | 97.6 |

still achieves robust results, such as 94.7% R@1 on Pitts30k-test, 96.9% R@1 on Pitts250k-test, and 98.1% R@1 on Tokyo24/7. These results confirm that the proposed Soft Probing module and the dynamic graph sampling strategies are inherently effective at enhancing feature representation and discriminability, even in the absence of explicit cross-image feature interaction.

Table 14: Evaluation results of SAGE (ViT-B and ViT-L) **without** the InteractHead at $322 \times 322$. Note that Nordland⋆ evaluates 2,760 summer queries against a 27,592 winter database, whereas Nordland uses 27,592 winter queries against a 27,592 summer database; see App. A.3 for full details.

| Method | Pitts30k-test | | | MSLS-val | | | Nordland | | | Tokyo24/7 | | | SPED | | |
|---|---|---|---|---|---|---|---|---|---|---|---|---|---|---|---|
| | R@1 | R@5 | R@10 | R@1 | R@5 | R@10 | R@1 | R@5 | R@10 | R@1 | R@5 | R@10 | R@1 | R@5 | R@10 |
| SAGE (-B) | 93.4 | 97.0 | 97.9 | 93.4 | 97.3 | 97.6 | 94.1 | 98.0 | 98.8 | 97.1 | 98.4 | 99.0 | 91.8 | 95.7 | 96.5 |
| SAGE (-L) | 94.7 | 97.7 | 98.4 | 94.2 | 97.8 | 98.1 | 94.8 | 98.2 | 98.9 | 98.1 | 98.1 | 98.7 | 92.1 | 96.2 | 96.9 |

| Method | Pitts250k-test | | | Nordland⋆ | | | Eynsham | | | SF-XL-Small | | | AmsterTime | | |
|---|---|---|---|---|---|---|---|---|---|---|---|---|---|---|---|
| | R@1 | R@5 | R@10 | R@1 | R@5 | R@10 | R@1 | R@5 | R@10 | R@1 | R@5 | R@10 | R@1 | R@5 | R@10 |
| SAGE (-B) | 95.7 | 98.6 | 99.2 | 86.7 | 95.9 | 97.0 | 92.3 | 95.8 | 96.5 | 88.8 | 90.5 | 90.9 | 63.1 | 82.5 | 86.2 |
| SAGE (-L) | 96.9 | 99.2 | 99.6 | 87.4 | 96.2 | 97.5 | 92.3 | 95.9 | 96.7 | 89.0 | 91.8 | 92.2 | 65.6 | 86.7 | 90.4 |

Beyond the quantitative ablations, Figure 9 illustrates representative qualitative failure modes of SAGE. Most incorrect top-1 results arise in highly ambiguous scenes, where the query and the retrieved image share strong low-level cues such as road layout, building facades, and vanishing-point geometry, yet correspond to different geographic places. As shown in the figure, these errors are concentrated in challenging conditions including severe viewpoint changes, strong illumination or weather shifts, long-term appearance changes, and dynamic occlusions caused by vehicles or pedestrians. In such cases, even though SoftP and the dynamic geo-visual sampling strategy improve the model's ability to emphasize discriminative local regions and to mine informative neighborhoods during training, the remaining failure cases suggest that the current descriptor can still be confused by near-duplicate urban structures or by transient visual evidence that dominates the scene. We believe future improvements may require stronger structural reasoning, more explicit handling of transient objects, or finer-grained local matching in the most ambiguous environments.

## A.3 DATASET DETAILS

**Pitts30k-test** (Torii et al., 2013). The Pitts30k-test dataset is a subset of the Pittsburgh 250k dataset collected from Google Street View panoramas with GPS labels. It contains urban street-view images from Pittsburgh, Pennsylvania, USA, covering diverse city environments such as roads, bridges, and buildings, with large variations in viewpoint, season, and illumination. Each location provides multiple viewpoint images, and the dataset is primarily used for evaluating VPR models in urban scenarios.

**SPED** (Chen et al., 2017). The SPED dataset, also known as the Specific PlacEs Dataset, consists of images captured by fixed surveillance cameras over extended time periods, covering significant changes in illumination, weather, and seasons. It contains unique locations, each with hundreds of images taken at different times of day and year, providing a challenging benchmark for long-term VPR under extreme appearance variations.

**MSLS-val** (Warburg et al., 2020). The MSLS-val dataset is the validation split of the Mapillary Street-Level Sequences dataset, which contains street-level imagery sequences from cities worldwide captured with various devices such as smartphones, dashcams, and professional mapping cameras. The validation set covers multiple cities, seasons, and weather conditions, and is widely used for tuning and evaluating VPR models under diverse geographic and environmental variations.

**Nordland** (Sünderhauf et al., 2013). The Nordland dataset consists of front-facing video recordings from a train journey along the same railway route in Norway, captured in all four seasons: spring, summer, autumn, and winter. The route and camera viewpoints are fixed, ensuring identical spatial structure while presenting extreme appearance changes due solely to seasonal and weather differences, making it a valuable benchmark for cross-season VPR research.

**Tokyo247** (Torii et al., 2015). The Tokyo247 dataset contains urban street-view images from Tokyo, Japan, primarily sourced from Google Street View panoramas with GPS labels, and is named for including both day and night imagery to represent long-term scene changes. It features dense metropolitan areas with tall buildings as well as some open spaces, offering large variations in viewpoint, illumination, and dynamic objects, and is used to evaluate VPR models in complex urban environments.

**AmsterTime** (Yildiz et al., 2022). The AmsterTime dataset is a time-lapse street-view dataset of Amsterdam, containing images of the same locations captured in different years, such as 2008 and 2014, using Google Street View panoramas. It reflects long-term changes in building facades and urban infrastructure, providing a benchmark for evaluating VPR models under temporal urban transformations.

**Pitts250k-test** (Torii et al., 2013). The Pitts250k-test dataset is the test split of the Pittsburgh 250k dataset, consisting of street-view images from Pittsburgh and surrounding areas, captured from Google Street View panoramas with GPS labels. It offers diverse urban scenes with dense road networks, buildings, and bridges, and serves as a large-scale benchmark for evaluating the scalability and robustness of VPR systems.

**Eynsham** (Cummins & Newman, 2010; Berton et al., 2022b). The Eynsham dataset is a GPS-synchronized street-view image sequence collected along approximately 35 km of driving routes in Eynsham, Oxfordshire, UK, using a vehicle-mounted camera. It provides continuous video frames with precise ground-truth positions, making it suitable for sequence-based place recognition and loop closure detection experiments.

**SF-XL** (Berton et al., 2022a). The San Francisco eXtra Large (SF-XL) dataset is a massive and dense new benchmark designed to push visual geo-localization research towards realistic, city-wide applications. Comprising over 41 million images captured across a decade, it presents significant real-world challenges such as long-term temporal variations and a domain shift between its Street View database and crowd-sourced queries.

**Nordland⋆** (Sünderhauf et al., 2013). The Nordland⋆ dataset introduces a challenging cross-seasonal evaluation protocol to test model robustness against extreme appearance shifts. Specifically, it uses $2,760$ summer images as queries against a complete winter sequence of $27,592$ reference images, rigorously evaluating the models' capability to handle severe environmental variations caused by seasonal changes.

## A.4 COMPARED METHODS DETAILS

**NetVLAD** (Arandjelovic et al., 2016)[1] CVPR' 2016. A classic VPR method with a learnable VLAD layer pluggable into any CNN. Uses VGG-16 backbone, trained on Pitts30k, optimized via weakly supervised ranking loss, outperforming traditional methods.

**SFRS** (Ge et al., 2020)[2] ECCV' 2020. Addresses GPS noise by mining hard positives via self-supervised fine-grained region similarities, multi-generation training. Based on NetVLAD, VGG-16 backbone, trained on Pitts30k, outperforming state-of-the-art then.

**CosPlace** (Berton et al., 2022a)[3] CVPR' 2022. Solves scalability in large-scale localization by framing training as classification. Constructs SF-XL dataset, uses CosPlace Groups. VGG-16/ResNet backbone, outputs 512D descriptors, low memory usage, suitable for city-scale applications.

**MixVPR** (Ali-Bey et al., 2023b)[4] WACV' 2023. This is a novel holistic feature aggregation method for VPR. It takes feature maps from pre-trained backbones as global features and iteratively incorporates global relationships into each feature map through stacked Feature-Mixer blocks (composed solely of multi-layer perceptrons), without the need for local or pyramidal aggregation. Using backbones like ResNet and trained on datasets such as GSV-Cities, it outperforms existing methods by a large margin with less than half the number of parameters compared to CosPlace and NetVLAD.

**R2Former** (Zhu et al., 2023)[5] CVPR' 2023. This method introduces R2Former, a unified framework that employs a pure Transformer architecture to handle both global retrieval and local reranking in a single, end-to-end model. Its novel reranking module replaces slow geometric verification with a learnable Transformer that analyzes richer cues like feature correlation and attention, achieving state-of-the-art accuracy while dramatically reducing inference time and memory usage.

**EigenPlaces** (Berton et al., 2023)[6] ICCV' 2023. This method embeds viewpoint robustness into learned global descriptors by clustering training data to explicitly present the model with different views of the same points of interest, without extra supervision. Using backbones like VGG-16 or ResNet with GeM pooling and trained on the SF-XL dataset, it outperforms state-of-the-art methods on most datasets, requiring 60% less GPU memory for training and using 50% smaller descriptors.

**SelaVPR** (Lu et al., 2024c)[7] ICLR' 2024. It proposes a hybrid global-local adaptation method that adapts pre-trained foundation models (e.g., DINOv2) via lightweight adapters without modifying the pre-trained model parameters, efficiently generating global features for candidate retrieval and local features for re-ranking. It also introduces a mutual nearest neighbor local feature loss to avoid time-consuming spatial verification. Outperforming state-of-the-art methods on benchmarks like MSLS, it consumes only about 3% of the retrieval time of RANSAC-based two-stage methods.

**CricaVPR** (Lu et al., 2024b)[8] CVPR' 2024. It proposes cross-image correlation-aware representation learning, using attention to correlate features of multiple images in a batch, enabling each image feature to gain useful information from others for enhanced robustness. A multi-scale convolution-enhanced adaptation method is designed to insert lightweight adapters into frozen pre-trained foundation models (e.g., DINOv2) to introduce multi-scale local information. It outperforms state-of-the-art methods by a large margin on multiple benchmarks with shorter training time.

**SALAD** (Izquierdo & Civera, 2024)[9] CVPR' 2024. It reformulates NetVLAD's soft assignment of local features to clusters as an optimal transport problem, considering both feature-to-cluster and cluster-to-feature relations, and introduces a 'dustbin' cluster to discard non-informative features (Xu et al., 2024). Using DINOv2 as the backbone with fine-tuning, it trains in only 4 epochs. This single-stage method outperforms both single-stage and two-stage methods, with fast inference speed.

---

[1]https://github.com/Nanne/pytorch-NetVlad

[2]https://github.com/yxgeee/OpenIBL

[3]https://github.com/gmberton/CosPlace

[4]https://github.com/amaralibey/MixVPR

[5]https://github.com/bytedance/R2Former

[6]https: //github.com/gmberton/EigenPlaces

[7]https://github.com/Lu-Feng/SelaVPR

[8]https://github.com/Lu-Feng/CricaVPR

[9]https://github.com/serizba/salad

**BoQ** (Ali-bey et al., 2024)[10] CVPR' 2024. This method learns a set of global queries and uses cross-attention to probe input features for consistent information aggregation. It supports both CNN and Vision Transformer backbones, trained on the GSV-Cities dataset. As a one-stage global retrieval method without re-ranking, surpasses two-stage methods, and is fast and efficient.

**SALAD-CM** (Izquierdo & Civera, 2025)[11] ECCV' 2024. This work addresses the insufficient Geographic Distance Sensitivity (GDS) of existing VPR models by proposing a novel sample mining strategy. It constructs a graph of visually similar images and samples cliques (sets of geographically close images) from the graph as training batches to enhance the model's ability to distinguish small-range geographic distances. Based on models like DINOv2 SALAD and MixVPR, trained on densely sampled datasets such as MSLS and Nordland, it significantly improves recall without increasing inference computational overhead.

**SuperVLAD** (Lu et al., 2024d)[12] NIPS' 2024. This method improves NetVLAD by removing cluster centers and using a small number of clusters , enhancing cross-domain generalization and simplifying the model. It also proposes 1-Cluster VLAD, which generates extremely low-dimensional descriptors by introducing "ghost clusters" and outperforms methods like GeM pooling with the same dimension. Using Transformer backbones (e.g., DINOv2) and trained on datasets like Pitts30k, it outperforms existing methods with lower feature dimensions.

**EMVP** (Qiu et al., 2024)[13] NIPS' 2024. This method leverages Visual Foundation Models(e.g., DINOv2) and proposes a Centroid-Free Probing (CFP) stage that uses second-order features to better adapt VFM descriptors. It introduces a Dynamic Power Normalization (DPN) module to adaptively preserve task-specific information in both recalibration and CFP stages, forming a Parameter Efficiency Fine-Tuning (PEFT) pipeline. It achieves excellent performance on datasets like MSLS and Pitts250k, saving $64.3\%$ trainable parameters compared to existing state-of-the-art PEFT methods.

**VLAD-BuFF** (Khaliq et al., 2024)[14] ECCV' 2024. This method improves VPR accuracy by implementing a burst-aware weighting mechanism that discounts repetitive features to emphasize more distinctive visual cues. Simultaneously, it achieves high computational efficiency by using a pre-projection layer for local features, initialized with PCA, which enables rapid aggregation in a lower-dimensional space while maintaining high recall.

**EDTFormer** (Jin et al., 2025b)[15] TCSVT' 2025. This method introduces EDTformer, a simplified transformer decoder architecture that utilizes a set of learnable queries to efficiently decode and aggregate crucial information from image features for robust place recognition. Furthermore, it enhances the DINOv2 backbone with a novel Low-rank Parallel Adaptation (LoPA) method, which enables highly memory and parameter-efficient fine-tuning to deliver high performance with lower training costs.

**EffoVPR** (Tzachor et al., 2025) ICLR' 2025. This method introduces EffoVPR, which effectively utilizes a foundation model by extracting powerful local descriptors from its internal self-attention layers for a highly effective re-ranking process, even in a zero-shot setting. Furthermore, its single-stage approach achieves state-of-the-art performance with exceptionally compact global features by simplifying training to fine-tune only the model's final layers, thus eliminating the need for external aggregation modules.

**FoL** (Wang et al., 2025)[16] AAAI' 2025. This two-stage VPR method models reliable discriminative regions via Extraction-Aggregation Spatial Alignment Loss (SAL) and Foreground-Background Contrast Enhancement Loss (CEL), guiding global feature generation and efficient re-ranking. It introduces a weakly supervised local feature training strategy based on pseudo-correspondences and a discriminative region-guided efficient re-ranking pipeline. Using DINOv2 as the backbone and trained on GSV-Cities, it outperforms existing two-stage methods on multiple benchmarks with higher computational efficiency.

---

[10]https://github.com/amaralibey/ Bag-of-Queries
[11]https://github.com/serizba/cliquemining
[12]https://github.com/lu-feng/SuperVLAD
[13]https://github.com/vincentqqb/EMVP
[14]https://github.com/Ahmedest61/VLAD-BuFF
[15]https://github.com/Tong-Jin01/EDTformer
[16]https://github.com/chenshunpeng/FoL

**MegaLoc** (Berton & Masone, 2025)[17] CVPRW' 2025. This method introduces a unified image retrieval model effective across VPR, Landmark Retrieval, and Visual Localization. By employing a DINO-v2 backbone and a SALAD aggregation layer, it effectively fuses five diverse datasets via tailored sampling strategies and utilizes a memory-efficient training approach to optimize a multi-similarity loss.

**ImAge** (Lu et al., 2025b)[18] NIPS' 2025. This method introduces an Implicit Aggregation paradigm for VPR that eliminates explicit aggregators. By prepending learnable aggregation tokens before a specific transformer block, it leverages the inherent self-attention mechanism to implicitly aggregate global context from patch tokens. The final representation is directly formed from these output tokens, which are initialized via K-means clustering for enhanced robustness.

**SelaVPR++** (Lu et al., 2025a)[19] TPAMI' 2025. This method introduces an efficient two-stage VPR paradigm. It adapts frozen foundation models using memory-efficient MultiConv adapters. For retrieval, it eliminates local feature matching by using compact binary features for fast initial search and floating-point global features for accurate re-ranking, all trained on a unified multi-dataset protocol.

### A.5 LIMITATIONS & DISCUSSIONS & FUTURE WORK

**Limitations and Discussions.**

While achieving strong performance, we identify a few areas for further exploration. First, navigating highly dynamic scenes with rapid occlusions (e.g., as shown in Fig. 9) remains an ongoing challenge, as reliable background cues may be temporarily obscured. Second, while the per-epoch overhead of Online Graph Creation is entirely acceptable on current benchmarks, scaling to extraordinarily large datasets might incur increased computational demands, representing a conscious balance between efficiency and adaptability. Third, our graph construction benefits from reasonably accurate geographic coordinates. In scenarios with highly noisy or sparse GPS data, learning granular spatial distinctions may require further robust adaptations.

**Future Work.**

First, to better handle dynamic scenes, future work could integrate foundation models like SAM and CLIP to explicitly mask transient objects, encouraging the model to focus on stable background contexts. Additionally, exploring multi-view 3D representations offers a potential direction. By reasoning across multiple viewpoints and aggregating geometry-grounded features, the framework could capture richer spatial structures to enhance generalization in complex environments.

Second, to enhance robustness against extreme domain shifts (e.g., historical vs. modern images), we plan to incorporate broader multimodal cues (Ou et al., 2025; Xu et al., 2025). Fusing visual features with textual metadata, 3D contextual reasoning, or topological graph structures could provide complementary and rich semantic signals for more resilient matching across varying domains.

Finally, the core principle of adaptive graph-based sampling holds promise beyond VPR. Applying this dynamic, "slow thinking" paradigm to other deep metric learning tasks, such as person re-identification or fine-grained image retrieval, could be a fruitful area of research.

---

[17]https://github.com/gmberton/megaloc
[18]https://github.com/Lu-Feng/ImAge
[19]https://github.com/Lu-Feng/SelaVPRplusplus

