# OpenReview forum: "SAGE: Spatial-visual Adaptive Graph Exploration for Efficient Visual Place Recognition"
_ICLR.cc/2026/Conference — ICLR 2026 Poster_

### Official Review · Reviewer_EQqS · 2025-10-28

**Soundness:** 3
**Presentation:** 3
**Contribution:** 4
**Rating:** 8
**Confidence:** 5

**Summary:**

This paper proposes SAGE (Spatial-Visual Adaptive Graph Exploration), a novel training framework for Visual Place Recognition (VPR) that dynamically integrates spatial and visual cues through online graph construction and adaptive sampling.
Unlike prior methods that use static sampling or fixed descriptor aggregation, SAGE introduces:
1. Soft Probing (SoftP): a lightweight, residual reweighting module that amplifies discriminative local patches before aggregation;
2. InteractHead: a cross-image attention block improving descriptor coherence across views;
3. Dynamic Geo-Visual Graph (OGC): rebuilt each epoch to reflect evolving embeddings;
4. Greedy Weighted Sampling (GWS): an adaptive clique-expansion strategy for efficient hard sample mining.

**Strengths:**

1. Novel Training Paradigm: The idea of reconstructing an online geo-visual graph per epoch is elegant and well-justified. It aligns with the dynamic nature of embedding spaces and resolves the “stale hard samples” issue common in previous static mining methods.
2. Effective Feature Enhancement: The SoftP module is a simple yet powerful mechanism that improves local discriminability with negligible computational overhead. The InteractHead effectively models cross-view consistency, leading to more coherent global descriptors.
3. Comprehensive Empirical Validation: Results across eight datasets convincingly show SAGE’s superiority over strong baselines such as EMVP (NeurIPS’24), FoL (AAAI’25), and SALAD-CM (ECCV’24). The method also performs remarkably well with compact descriptors (4096-D), highlighting scalability.
4. Parameter Efficiency: Freezing DINOv2 while introducing only lightweight trainable modules (SoftP, DPN, InteractHead) demonstrates impressive efficiency—achieving SOTA results with fewer parameters than most competing methods.
5. Clear Presentation and Strong Technical Depth: The paper is well-written, logically organized, and provides strong ablation and visualization support (e.g., t-SNE clustering, SoftP heatmaps, AID analysis).

**Weaknesses:**

Although efficient in parameters, the online graph construction and clique expansion may increase runtime. A small ablation or runtime comparison would strengthen the claim of overall efficiency.

**Questions:**

None.

---

> ### Author Response · Authors · 2025-11-21
> **Response to Reviewer EQqS**
>
> Thanks for your positive recommendation and valuable suggestions. The following are Responses to the Weaknesses.
>
> **W1. Ablation of Component Runtime**
>
> Thank you for your valuable feedback on the training efficiency of our method. We have conducted a supplementary experiment to show the time cost and performance of online versus offline Graph Creation strategies. The results are as follows:
>
> **Table A: Online Graph Creation**
> | Method | Mining (min) | Train (min) | Total (min) | SPED (R@1/5/10) | MSLS-val (R@1/5/10) |
> | :--- | :---: | :---: | :---: | :---: | :---: |
> | cliquemining | 4.3 | 25.1 | 29.4 | 90.0 / 95.4 / 96.2 | 94.3 / 96.9 / 97.6 |
> | SAGE (w/o GWS) | 6.1 | 28.4 | 34.5 | 96.8 / 98.2 / 98.7 | 93.6 / 96.8 / 97.1 |
> | SAGE (full) | 6.2 | 28.4 | 34.6 | **98.9** / **99.7** / **100** | **94.5** / **97.4** / **97.8** |
>
> **Table B: Offline Graph Creation**
> | Method | Mining (one-time, min) | Train / Epoch (min) | SPED (R@1/5/10) | MSLS-val (R@1/5/10) |
> | :--- | :---: | :---: | :---: | :---: |
> | cliquemining | 21.6 | 25.1 | 89.5 / 94.9 / 96.1 | 94.2 / 97.2 / 97.4 |
> | SAGE (w/o GWS) | 30.7 | 28.4 | 96.7 / 98.2 / 98.9 | 93.5 / 96.6 / 97.1 |
> | SAGE (full) | 30.9 | 28.4 | **98.5** / **99.3** / **99.5** | **94.2** / **97.3** / **97.7** |
>
> *Note: Time in Table A is per epoch (mining 200 batches). The mining time in Table B is a one-time cost from an offline process (mining 1000 batches).* The experimental data indicates:
>
> 1.  Online Creation is a Cost-Effective Trade-off: While our online strategy (Table A) leads to an approximate 17.7% increase in training time, this is **a necessary cost for the significant performance gains achieved**. When comparing online and offline strategies (Table B), the online SAGE model demonstrates superior performance on key metrics such as SPED and MSLS-val. This validates the core argument of our paper: dynamic mining continuously captures the hard negatives that are most valuable to the model at its current state, thereby breaking through the performance bottlenecks of static mining.
>
> 2.  The Core Component is Highly Efficient: Our core component, GWS (Greedy Weighted Sampling), introduces a negligible 0.1-minute increase in training time while delivering substantial performance gains (e.g., boosting SPED R@1 from 96.8% to 98.9%). This proves its lightweight and efficient design.
>
> 3.  Overhead is Limited to the Training Phase: We wish to re-emphasize that the aforementioned overhead is confined to the training phase only. The inference efficiency of our model is on par with that of similar single-stage methods, with **no additional computational burden**.
>
> Thanks again for your thoughtful review, and please let us know if you have further questions.

---

### Official Review · Reviewer_AM9N · 2025-10-28

**Soundness:** 2
**Presentation:** 1
**Contribution:** 2
**Rating:** 2
**Confidence:** 4

**Summary:**

The paper presents a new model for Visual Place Recognition (VPR), which adds many different improvements on top of existing methods. Results show consistent improvements over existing methods, although it is not clear if the evaluation settings are fair towards other VPR methods.

**Strengths:**

Good results
Thorough evaluation against other methods

**Weaknesses:**

1. Some text is obviously hallucinated. SPED stands for Specific PlacEs Dataset, not Street View Place Recognition Dataset (line 894). Tokyo247 is called 24/7 because it has night imagery (24 hours), not because it contains approximately 247 km, as written in line 909.

2. There are hallucination-like unsubstantiated claims throughout the paper, some of which are quite arguable at best, like
- line 052: "However, approaches driven by VFMs or learned queries often demand careful regularization and substantial resources in terms of computation and data to ensure stable fine-tuning." What is this referring to? I'm not aware of any paper supporting this claim, and models like SALAD and BoQ (DINOv2-based) do not demand careful regularization and substantial resources, but quite the opposite.
- line 078: "This renders them less effective in scenarios that demand awareness of spatial context and the evolving appearance of hard samples". This is simply false, methods based on VFMs are more effective in every scenario, in fact SOTA VPR methods since 2024 all rely on VFMs, as proved by literally all methods on table 3.
- line 320: "Since the EMVP repository is actively maintained". EMVP codebase is unavailable, and the link indicated by EMVP authors returns 404 (https://github.com/vincentqqb/EMVP)

3. The method is overall quite hard to understand, which makes it difficult to evaluate its soundness and novelty. Figure 2 does not help in understanding. For example, line 241 suggests that B images form a sequence, and the sequence is processed as one, meaning that the descriptor of an image are influence by the descriptors of other images in the batch. It is not clear if this happening also at test time? If yes, this is a significant (unfair) advantage against other baselines, because it relies on the locality of queries (as consecutive queries in many datasets come from the same areas), whereas other methods aim to localize queries individually.

4. The paper would be significantly stronger if results were also shown on SF-XL, which is the only city-wide VPR dataset to date. Also, SF-XL (or even SF-small) does not have consecutive / nearby queries so it would lift doubts about evaluation fairness.

5. Line 112 is contradictory: what does it mean "tuning a frozen DINOv2 backbone"? Is it tuned or frozen? Later on PEFT is mentioned making it look like the actual DINOv2 backbone is frozen but LoRA is used

6. PEFT is not cited anywhere. PEFT should be cited, as well as the LoRA paper if a LoRA is used.

7. The OGC seems very similar to SALAD-CM graph, with the only difference that SALAD-CM creates it only once while OGC is created multiple times. Are there any other differences?

8. The ablation is incomplete. Given that the paper mentions that online graph creation is necessary, while previous methods compute it offline (like SALAD-CM using SALAD to compute graph only once at the beginning) the ablation should present results with offline graph creation as well.

9. The value of the threshold τ (line 259) is never defined.

10. The paper claims that Dynamic Geo-Visual Graph Mining leads to faster convergence (line 107 and line 293), but this is not proven nor discussed anywhere.

11. Eynsham is a dataset created in the paper "Deep Visual Geo-localization Benchmark", which must be cited, using raw data by Cummins & Newman. Also, Eynsham uses a 25 threshold, not 5 meters (line 315).

12. OGC is sometimes defined as Online Graph Creation and sometimes as Online Graph Construction (minor weakness).

**Questions:**

See weaknesses above

---

> ### Author Response · Authors · 2025-11-21
> **Response to Reviewer AM9N (Part 1/5)**
>
> Thanks for your comments to help us improve the paper. The following are Responses to the Questions/Weaknesses.
>
> **W1. On the Naming of Datasets in the Appendix**
>
> You are correct regarding the origins of the dataset names. We apologize for this oversight and have revised the appendix as follows:
>
> *   The explanation for "SPED" has been corrected to "Specific PlacEs Dataset" [1].
> *   The origin of the name "Tokyo247" has been corrected to reflect that it includes both day and night imagery (representing long-term scene changes across seasons and years, for all-weather, all-time place recognition) [2].
>
> **W2. On the Disputed Claims**
>
> We understand your concerns regarding several statements and acknowledge that our original phrasing was overly broad and could have led to misinterpretation. **Our intention was never to diminish the significant success of Visual Foundation Models (VFMs)**; on the contrary, our SAGE method is built upon a powerful VFM (DINOv2). We would like to clarify our original intent here.
>
> **Regarding Line 052 (Resource Usage):**
>
> We fully agree that methods like SALAD and BoQ, which use advanced fine-tuning paradigms, are highly efficient in their training process. Our original intent was to point out that different VFM adaptation strategies exist, with varying resource demands. For example, some methods fine-tune the last few Transformer encoder blocks of DINOv2 to boost performance. While effective, this approach **involves more trainable parameters and greater computational overhead compared to the Parameter-Efficient Fine-Tuning (PEFT) strategy** we employ in SAGE.
>
> This observation is also supported by recent research. For instance, EDTFormer (TCSVT 2025) [3] notes when discussing VFM adaptation strategies:
>
> > ...some research...began to enhance the performance of DINOv2 in VPR through only fine-tuning the last few encoder blocks... Unfortunately, this process is accompanied by a large number of trainable parameters. Meanwhile, CricaVPR and SelaVPR attempted to apply parameter efficient transfer learning (PETL) methods... However, this approach is efficient only in terms of parameters, not in training time and memory usage, as the gradient computation... still requires backpropagation through the pre-trained backbone...
>
> This supports our point about the trade-offs in resource efficiency among different VFM adaptation strategies.
> **We have revised the sentences** in the Introduction to: "While recent adaptation methods for VFMs are notably efficient, different strategies within the broader VFMs ecosystem vary in resource consumption [3]. For instance, fine-tuning a backbone's encoder layers typically requires more computational resources than Parameter-Efficient Fine-Tuning (PEFT) approaches [4,5]."
>
> **Regarding Line 078 (Hard Sample Mining):**
>
> Thank you for your comment. To clarify, our argument is not a critique of the capabilities of VFMs themselves, but rather an exploration of a training paradigm that can further unlock their potential. Specifically, we observed that many leading methods, while leveraging powerful VFM features, often rely on static or pre-defined sampling strategies. A limitation of such static approaches is their inability to adapt to the model's evolving understanding of "hard samples" during training. As the model improves, what was once "hard" may become easy, and new, more challenging examples emerge at the decision boundary.
> **We have rewritten the sentence** in the Introduction as follows: "Furthermore, while the model's understanding is dynamic, a pre-defined or static sampling policy [6] may fail to consistently present the most informative examples as learning progresses."
>
> **Regarding Line 320 (Baseline not available):**
>
> Thank you for your comment. Regarding the EMVP [4] repository, the link is indeed currently inaccessible. At the time of our experiments, the repository was available, and we based our implementation on the descriptions in their paper to ensure a fair comparison.
> **We have revised the sentence** as follows: "SAGE is built upon the EMVP framework, which we reproduced from its publication for a fair comparison as the official code is unavailable."
> Furthermore, to facilitate future research, we will make our codebase and models publicly available.
>
> **References**
> ```
> [1] Chen, Zetao, et al. Deep learning features at scale for visual place recognition. ICRA 2017.
> [2] Torii, Akihiko, et al. 24/7 place recognition by view synthesis. CVPR 2015.
> [3] Jin, Tong, et al. EDTformer: An Efficient Decoder Transformer for Visual Place Recognition. TCSVT 2025.
> [4] Qiu, Qibo, et al. Emvp: Embracing visual foundation model for visual place recognition with centroid-free probing. NIPS 2024.
> [5] Jia, Menglin, et al. Visual prompt tuning. ECCV 2022.
> [6] Izquierdo, Sergio, and Javier Civera. Close, but not there: Boosting geographic distance sensitivity in visual place recognition. ECCV 2024.
> ```

---

> ### Author Response · Authors · 2025-11-21
> **Response to Reviewer AM9N (Part 2/5)**
>
> **W3. Clarification on the Proposed Method and Figure 2**
>
> We thank the reviewer for pointing out the need for a clearer explanation of our method. Our SAGE framework addresses key challenges in VPR through several synergistic modules:
>
> *   The SoftP Feature Interaction part aims to **enhance the discriminability and robustness of image descriptors**. It consists of two lightweight components: SoftP (Fig. 2d) amplifies information-rich local features through data-driven residual weighting, while the InteractHead (part of Fig. 2a) establishes relationships among images within a batch to learn a more generalizable global feature representation.
> *   The Dynamic Geo-Visual Graph Mining is the core of SAGE's training strategy. We abandon static sampling policies. Instead, in each training epoch, we perform Online Graph Creation (Fig. 2b) to build a graph that fuses geographic proximity with the current visual similarity. Subsequently, we use Greedy Weighted Sampling (GWS, Fig. 2c) to mine the most challenging sample clusters from this graph for training. The design of Figure 2 is intended to intuitively illustrate this training workflow.
> *   Our InteractHead module is utilized during both training and testing phases. This design follows the established evaluation paradigm in recent top-tier VPR research. Specifically, our evaluation protocol is consistent with leading works such as CricaVPR (CVPR 2024) [1] and SuperVLAD (NIPS 2024) [2], which also apply similar cross-image information interaction mechanisms during both training and testing. The purpose of this design is not to exploit the "locality of queries" within a specific dataset, but rather to **enhance feature discriminability by modeling the correlations between images**. Our subsequent experimental results on the SF-small benchmark further validate this point.
>
> **W4. Evaluation on SF-XL Dataset and Fairness Verification**
>
> We agree with the reviewer that testing on a dataset where queries are not consecutive is crucial for verifying a method's generalization ability and ensuring a fair evaluation. We have conducted comprehensive experiments on the SF-small benchmark. SF-small is derived from the large-scale SF-XL dataset [3] and similarly **does not contain consecutive or neighboring queries**. Our experimental results are as follows:
>
> | Method | SF-small | | |
> | :--- | :--- | :--- | :--- |
> | | **R@1** | **R@5** | **R@10** |
> | CricaVPR (CVPR 2024) | 84.5 | 88.7 | 89.2 |
> | SALAD (CVPR 2024) | 85.7 | 88.2 | 89.7 |
> | SuperVLAD (NIPS 2024) | 85.8 | 89.1 | 89.5 |
> | SALAD-CM (ECCV 2024) | 84.0 | 88.0 | 89.8 |
> | EDTformer (TCSVT 2025) | 87.9 | 89.8 | 90.6 |
> | EMVP (NIPS 2024) | 88.2 | 90.6 | 91.1 |
> | **SAGE (Ours)** | **89.3** | **91.5** | **91.9** |
>
> As shown in the table, SAGE achieves 89.3% R@1 on SF-small, significantly outperforming all compared methods. This result strongly demonstrates that our method's performance gain stems from its advanced feature enhancement and dynamic mining strategies, rather than from exploiting any specific dataset biases.
>
> Furthermore, this result leads to an interesting observation: the performance of SALAD-CM [4] is lower than its baseline, SALAD [5]. This can be explained by the limitations discussed in the SALAD-CM paper itself, where the authors state that the effectiveness of their CliqueMining strategy is highly dependent on the presence of a Geographic Distance Sensitivity (GDS) problem in the dataset. On a dataset like SF-XL [3], where the primary challenge is viewpoint variation, this strategy may not yield positive results [4].
>
> In contrast, SAGE achieves a significant performance improvement over the strong EMVP baseline, highlighting our method's generality and robustness in addressing diverse challenges. We believe that the above clarifications and our **SOTA experimental results on SF-small have fully addressed your concerns** regarding methodological clarity, evaluation fairness, and generalization ability.
>
> **References**
> ```
> [1] Lu, Feng, et al. Cricavpr: Cross-image correlation-aware representation learning for visual place recognition. CVPR 2024.
> [2] Lu, Feng, et al. SuperVLAD: Compact and robust image descriptors for visual place recognition. NIPS 2024.
> [3] Berton, Gabriele, Carlo Masone, and Barbara Caputo. Rethinking visual geo-localization for large-scale applications. CVPR 2022.
> [4] Izquierdo, Sergio, and Javier Civera. Close, but not there: Boosting geographic distance sensitivity in visual place recognition. ECCV 2024.
> [5] Izquierdo, Sergio, and Javier Civera. Optimal transport aggregation for visual place recognition. CVPR 2024.
> ```

---

> ### Author Response · Authors · 2025-11-21
> **Response to Reviewer AM9N (Part 3/5)**
>
> **W5. Regarding the DINOv2 Backbone and LoRA**
>
> Thank you for your valuable feedback. We understand that the phrase "tuning a frozen ... backbone" could be confusing upon literal interpretation. We would like to clarify that in our method, all pretrained weights of the DINOv2 backbone are indeed completely frozen during training and undergo no updates.
>
> The "tuning" we refer to is achieved through the strategy of Parameter-Efficient Fine-Tuning (PEFT). Specifically, we **do not fine-tune the parameters of DINOv2 itself**. Instead, we insert lightweight, learnable Dynamic Power Normalization (DPN) layers into its frozen encoder modules [1].
>
> To resolve this ambiguity, we will revise the statement on line 112 in the revised version of our paper from:
>
> > "By efficiently tuning a frozen DINOv2 backbone..."
>
> to the more precise:
>
> > "Implemented with a frozen DINOv2 backbone and parameter-efficient fine-tuning..."
>
> This revision clarifies our technical approach: the backbone is frozen, and the fine-tuning is applied to the added, parameter-efficient modules.
>
> **W6. Regarding Citations for PEFT and LoRA**
>
> Thank you for pointing this out. Our method falls within the scope of PEFT [2,3], but as we adopt a DPN strategy [1], **we do not use the LoRA** [4]. This is the reason why the LoRA paper was not cited.
>
> Furthermore, you are correct in pointing out that we did not provide a formal citation for the general concept of PEFT. This was an oversight. To address this, we will add the following introduction to the Related Work section of our paper to properly describe and cite PEFT:
>
> > "To adapt large Visual Foundation Models (VFMs) for downstream tasks, Parameter-Efficient Fine-Tuning (PEFT) has emerged as a crucial paradigm [2,3,4]. These methods focus on updating only a small subset of model parameters rather than the entire backbone, significantly enhancing training efficiency while mitigating overfitting."
>
> This added discussion will help readers better understand the technical context of our work.
>
> **W7. The Distinction Between OGC and SALAD-CM**
>
> Thank you for raising this question, as it gives us an opportunity to clarify the core innovations of our Online Graph Creation (OGC) method. Beyond simply being "created multiple times," OGC fundamentally differs from SALAD-CM's Graph Creation [5] in the following two key aspects:
>
> 1.  Graph Creation Based on "Evolving" Visual Features: The core philosophy of SAGE is that an effective sample mining strategy must be synchronized with the model's own learning progress. Our OGC recalculates the visual similarity between nodes in every epoch using the model's current embedding representations. This means that as the model's discriminative power improves, the graph's topology "evolves" accordingly, enabling it to continuously provide challenging samples to the model.
>
> 2.  Fused Geo-Visual Edge Weighting: Our method directly defines the graph's edges using a unified affinity score. Integrating geographic proximity and visual similarity into a single metric allows the Graph Creation to balance both types of information simultaneously. In contrast, SALAD-CM, as described, employs what is more akin to a two-step strategy: it first constructs a graph based on visual similarity, and then samples geographically close cliques from that graph.
>
> In summary, OGC is not merely a "repeated" static process, but rather a dynamic and adaptive process that is tightly coupled with the model's evolving learning state.
>
> **References**
> ```
> [1] Qiu, Qibo, et al. Emvp: Embracing visual foundation model for visual place recognition with centroid-free probing. NIPS 2024.
> [2] Houlsby, Neil, et al. Parameter-efficient transfer learning for NLP. ICML 2019.
> [3] Jia, Menglin, et al. Visual prompt tuning. ECCV 2022.
> [4] Hu, Edward J., et al. Lora: Low-rank adaptation of large language models. ICLR 2022.
> [5] Izquierdo, Sergio, and Javier Civera. Close, but not there: Boosting geographic distance sensitivity in visual place recognition. ECCV 2024.
> ```

---

> ### Author Response · Authors · 2025-11-21
> **Response to Reviewer AM9N (Part 4/5)**
>
> **W8. Ablation Study on Offline Graph Creation**
>
> This is an excellent suggestion. Following your recommendation, we have **conducted a new set of ablation studies**, focusing on evaluating the performance of the offline graph creation strategy you mentioned. The offline strategy involves building the graph once before the training begins, from which we mined 1000 batches to ensure sufficient initial sampling quality. Its performance and efficiency data are presented in the table below:
>
> **Table A: Offline Graph Creation**
> | Method | Mining (one-time, min) | Train / Epoch (min) | SPED (R@1/5/10) | MSLS-val (R@1/5/10) |
> | :--- | :---: | :---: | :---: | :---: |
> | cliquemining | 21.6 | 25.1 | 89.5 / 94.9 / 96.1 | 94.2 / 97.2 / 97.4 |
> | SAGE (w/o GWS) | 30.7 | 28.4 | 96.7 / 98.2 / 98.9 | 93.5 / 96.6 / 97.1 |
> | SAGE (full) | 30.9 | 28.4 | 98.5 / 99.3 / 99.5 | 94.2 / 97.3 / 97.7 |
>
> As shown in Table A, even with a static, offline graph, our SAGE framework still achieves highly competitive results (e.g., reaching 98.5% R@1 on SPED). We then further compared this with our proposed online strategy. The online strategy dynamically reconstructs the graph in each training epoch, mining 200 batches each time. The results are as follows:
>
> **Table B: Online Graph Creation**
> | Method | Mining (min) | Train (min) | Total (min) | SPED (R@1/5/10) | MSLS-val (R@1/5/10) |
> | :--- | :---: | :---: | :---: | :---: | :---: |
> | cliquemining | 4.3 | 25.1 | 29.4 | 90.0 / 95.4 / 96.2 | 94.3 / 96.9 / 97.6 |
> | SAGE (w/o GWS) | 6.1 | 28.4 | 34.5 | 96.8 / 98.2 / 98.7 | 93.6 / 96.8 / 97.1 |
> | SAGE (full) | 6.2 | 28.4 | 34.6 | 98.9 / 99.7 / 100 | 94.5 / 97.4 / 97.8 |
>
> By directly comparing the full SAGE performance under both strategies, we observe that online graph creation improves the R@1 performance on SPED from 98.5% to 98.9%, and on MSLS-val from 94.2% to 94.5%. This consistent performance improvement validates our hypothesis: a static, offline graph cannot fully adapt to the model's continuously evolving feature space during training, whereas a dynamic, online graph can consistently provide the model with the most challenging samples, thereby pushing the model's performance to a higher level.
>
> **W9. Definition of Thresholds**
>
> Thank you very much for pointing out this oversight. In our method, there are two hyperparameter thresholds, $ \tau_1 $ and $ \tau_2 $, which were not explicitly defined in the original manuscript. We apologize for this omission.
>
> *   $ \tau_1 $ is the geographic distance threshold used to initially filter candidate neighborhoods.
> *   $ \tau_2 $ is the affinity score threshold used to construct the final sparse graph from the weighted graph.
>
> We conducted a sensitivity analysis on these hyperparameters. The results of the ablation study are as follows:
>
> **Table C: Hyperparameter Sensitivity Analysis**
> | $ \tau_1 $ | MSLS-val (R@1/5/10) | $ \tau_2 $ | MSLS-val (R@1/5/10) |
> | :---: | :---: | :---: | :---: |
> | 20 | 93.8 / 97.3 / 97.7 | $ -2.75 \times 10^3 $ | 94.1 / 97.6 / 97.8 |
> | 25 | 94.5 / 97.4 / 97.8 | $ -2.88 \times 10^3 $ | 94.5 / 97.4 / 97.8 |
> | 30 | 93.7 / 97.0 / 97.4 | $ -3.00 \times 10^3 $ | 93.9 / 97.3 / 97.6 |
>
> The analysis of the experimental results is as follows:
> 1.  Parameter Selection: We chose $ \tau_1 = 25 $ and $ \tau_2 = -2.88 \times 10^3 $ in our experiments. As shown in the table, this set of parameters achieved the best overall performance on MSLS-val.
> 2.  Robustness: The experimental data also shows that the model's performance remains stable in the vicinity of these optimal values, without drastic fluctuations. This demonstrates that our method is not overly sensitive to these key hyperparameters and exhibits good robustness.

---

> ### Author Response · Authors · 2025-11-21
> **Response to Reviewer AM9N (Part 5/5)**
>
> **W10. Regarding Faster Convergence**
>
> When we mention "faster convergence" in our paper, we are referring to our method's ability to help the model reach a higher performance level in fewer training epochs, which directly reflects a **greater learning efficiency per epoch**. We conducted a new experiment specifically designed to measure the convergence efficiency of our method. We directly compared our full SAGE model against a variant that uses the Cliquemining (CM) sampling strategy [1]. We tracked the performance of both models during the early stages of training, and the results are as follows:
>
> **Table: Epoch-wise Performance on MSLS-val (R@1/5/10)**
> | Epoch | SAGE (w/ CM) | SAGE (Ours) |
> | :---: | :---: | :---: |
> | 2 | 92.3 / 96.1 / 96.6 | 92.5 / 96.5 / 97.1 |
> | 4 | 92.7 / 96.6 / 97.0 | 93.4 / 96.9 / 97.4 |
>
> The experimental results show that, firstly, by the 4th epoch, our full SAGE model already demonstrates a clear performance advantage (93.4% vs. 92.7% R@1). Secondly, the performance gap between our method and the CM variant widens from epoch 2 to epoch 4. This directly demonstrates that our dynamic sampling strategy guides the model towards more sustained and efficient learning, thereby achieving superior performance within the same number of training epochs.
>
> We thank you for this comment. We will add this table and the corresponding analysis to the revised version of our paper.
>
> **W11. Regarding the Eynsham Dataset Citation and Evaluation Threshold**
>
> Thank you for the suggestion. To provide a more comprehensive background on the Eynsham dataset [2], it is indeed necessary to add a citation to the "Deep Visual Geo-localization Benchmark" [3]. We have added this citation in Section 4.1 of our paper.
>
> Our experiments on Eynsham were conducted using the correct 25-meter threshold; the 5 meters mentioned in the text was an error made during writing. We apologize for any confusion this error may have caused and have **corrected it to 25 meters** in the text (line 315) to ensure it is fully consistent with our experimental setup.
>
> **W12. Regarding the Definitional Consistency of the Abbreviation OGC**
>
> We also thank you for highlighting the issue of the inconsistent definition of the abbreviation OGC in the paper. To ensure the clarity of the paper, we have standardized all related expressions to "Online Graph Creation" and have **reviewed the entire manuscript to ensure the consistency** of this term.
>
> Thank you again for the great efforts and valuable comments.
>
> **References**
> ```
> [1] Izquierdo, Sergio, and Javier Civera. Close, but not there: Boosting geographic distance sensitivity in visual place recognition. ECCV 2024.
> [2] Cummins, Mark, and Paul Newman. Highly scalable appearance-only SLAM–FAB-MAP 2.0. (2010).
> [3] Berton, Gabriele, et al. Deep visual geo-localization benchmark. CVPR 2022.
> ```

---

> > ### Comment · Reviewer_AM9N · 2025-11-24
> > **Can't recommend acceptance for a paper with many LLM hallucinations**
> >
> > My main concern is that a lot of text is obviously LLM generated and full of hallucinations (w1, w2, w11). While I did catch a few obvious ones, it is hard to know how many hallucinated statements there are. This is a grave issue that can propagate false information. While the authors claim to have fixed some of those statements (although the paper has not been updated), the lack of transparency and failure to recognize the issue that the text contains LLM hallucinations is concerning, hence this issue alone is enough to not recommend the paper for acceptance.
> >
> > Also, the paper has not been updated, but even that wouldn't fix the issue mentioned above.
> >
> > Moreover, the rebuttal is overly verbose, for example in w6 the authors used around 150 words just to say that PEFT will be cited.

---

> > > ### Author Response · Authors · 2025-11-25
> > >
> > > Dear Reviewer AM9N,
> > >
> > > Thank you for your timely feedback. **We have updated the paper PDF** and are grateful for your valuable suggestions.
> > >
> > > Please note that in response to Question 4 from Reviewer NcXJ, we have added "Failure cases of SAGE" in Figure 6 (line 486 of the main text). This took some additional time, and we apologize for any delay.
> > >
> > > We have provided detailed point-by-point responses to your suggestions. We apologize if these are lengthy, but we wanted to address your points thoroughly and confirm they were not generated by an LLM. The corresponding changes have been incorporated into the main text.
> > >
> > > For your convenience, here is a brief summary of the specific locations of our revisions:
> > >
> > > 1.  SPED and Tokyo247 datasets: Descriptions have been revised in the Appendix on lines 961 and 976.
> > > 2.  Resource Usage (previously Line 052): Addressed and revised on line 052, with an added citation for PEFT.
> > > 3.  Hard Sample Mining (previously Line 078): The explanation has been revised on line 078.
> > > 4.  Baseline not available (previously Line 320): Addressed on line 364.
> > > 5.  SF-XL dataset: A citation and results have been added to the Appendix on line 935, including a new comparison in Table 11.
> > > 6.  DINOv2 backbone clarification (previously Line 112): The clarification has been made on line 112.
> > > 7.  PEFT citation: Also added to the Related Work section on line 138.
> > > 8.  Offline graph creation ablation: We have added Table 7 on line 502 to present these results.
> > > 9.  Threshold τ value: The value has been added on line 375.
> > > 10. Convergence analysis: We have added Table 8 on line 515 to present these results.
> > > 11. Eynsham citation: An additional citation has been added in the main text (line 311) and the Appendix (line 992).
> > > 12. Terminology: We have globally replaced "Online Graph Construction" with "Online Graph Creation" for consistency.
> > >
> > > As the ICLR review process supports an interactive discussion and revisions, we are committed to continuously improving our manuscript and welcome any further feedback you may have.

---

### Official Review · Reviewer_NcXJ · 2025-10-31

**Soundness:** 2
**Presentation:** 2
**Contribution:** 2
**Rating:** 4
**Confidence:** 4

**Summary:**

This paper introduces SAGE , a new framework for VPR that addresses several challenges in large-scale geotagged image retrieval. VPR is essential for tasks such as autonomous robot navigation, loop closure detection in autonomous driving, and map construction. Previous methods have focused on fine-tuning descriptors or static sampling strategies but have failed to account for the dynamic interplay between spatial context and visual similarity during training. SAGE improves VPR performance by dynamically constructing an online geo-visual graph, enhancing local feature aggregation, and using hard sample mining.

**Strengths:**

1. SAGE introduces a dynamic approach to hard sample mining and geo-visual graph construction, addressing limitations in previous methods.
2. SAGE achieves top performance on multiple VPR benchmarks, demonstrating its effectiveness across various environmental conditions, such as viewpoint shifts, weather changes, and lighting variations.

**Weaknesses:**

1. The writing is not clear enough and fails to highlight the key contributions of this paper.
2. While the paper introduces the SAGE framework and its various components (such as SoftP, InteractHead, and the dynamic geo-visual graph mining), the justification for why these specific techniques were chosen over others is not sufficiently detailed. For instance, the paper does not provide a detailed comparison of SAGE’s components with other potential alternatives that might achieve similar outcomes.
3. The paper may benefit from a deeper analysis of why SoftP, InteractHead, and other components were selected. A discussion on related work and the performance of other possible methods in a similar context could enhance the novelty of the approach. For example, why are these methods preferable over existing techniques like spatial attention, or other forms of feature aggregation?
4. There is a mention of a performance drop in highly dynamic environments, but this is not sufficiently examined.
5. Please include more comparisons with the latest state-of-the-art methods.

**Questions:**

1. The writing is not clear enough and fails to highlight the key contributions of this paper.
2. While the paper introduces the SAGE framework and its various components (such as SoftP, InteractHead, and the dynamic geo-visual graph mining), the justification for why these specific techniques were chosen over others is not sufficiently detailed. For instance, the paper does not provide a detailed comparison of SAGE’s components with other potential alternatives that might achieve similar outcomes.
3. The paper may benefit from a deeper analysis of why SoftP, InteractHead, and other components were selected. A discussion on related work and the performance of other possible methods in a similar context could enhance the novelty of the approach. For example, why are these methods preferable over existing techniques like spatial attention, or other forms of feature aggregation?
4. There is a mention of a performance drop in highly dynamic environments, but this is not sufficiently examined.
5. Please include more comparisons with the latest state-of-the-art methods.

---

> ### Author Response · Authors · 2025-11-21
> **Response to Reviewer NcXJ (Part 1/3)**
>
> Thanks for your insightful comments and valuable suggestions. The following are Responses to the Questions/Weaknesses.
>
> **W1. Regarding the Highlighting of Main Contributions**
>
> Thank you very much for your valuable time and insightful comments on our paper. We appreciate this opportunity to more clearly articulate the key contributions of our work and to address your concerns.
>
> The core idea of our work is to introduce a "dynamic adaptive" training paradigm to address the hard sample mining problem in Visual Place Recognition (VPR). This is fundamentally different from previous methods that rely on static or predefined sampling strategies [1]. To address your concerns more clearly, we would like to reiterate and emphasize our key contributions:
>
> **1. Core Paradigm Innovation: Dynamic Geo-Visual Graph Mining**
>
> Traditional VPR methods typically perform a one-time clustering or grouping of hard samples based on their initial features. This approach ignores the critical fact that the **model's own embedding space is constantly evolving** during training.
>
> Our key innovation lies in an Online Graph Creation mechanism. In every training epoch, we **dynamically reconstruct an affinity graph** using the model's most current visual feature embeddings and geo-spatial information. This means our sampling strategy remains synchronized with the model's latest "learning state," enabling it to dynamically capture the most challenging samples at the current decision boundary.
>
> **2. Synergistic Techniques to Maximize the Paradigm's Effectiveness**
>
> To maximize the impact of the dynamic paradigm described above, we designed two key techniques that work in concert:
>
> *   SoftP for Feature Interaction: High-quality feature representations are crucial before Graph Creation and sampling can be effective. Our SoftP module provides a high-quality input for this dynamic mining by learning to enhance the discriminability of local patches in a data-driven manner, with a negligible parameter increase. Additionally, the InteractHead enables the model to learn common features across different images [2,3].
> *   Weighted Greedy Clique Expansion: This sampling algorithm precisely locates and extracts the most confusing sample clusters (cliques) from the dynamic graph for training, enabling highly efficient hard sample mining.
>
> **3. Demonstrated Effectiveness: SOTA Performance with High Efficiency**
>
> Efficiency: As shown in Table 4 of our paper, by freezing the backbone and Parameter-Efficient Fine-Tuning (PEFT) [4,5], SAGE achieves excellent performance while maintaining an **extremely low number of trainable parameters**, demonstrating exceptional parameter efficiency.
>
> SOTA Performance: As shown in Tables 2 and 3, SAGE **comprehensively surpasses previous SOTA methods on eight mainstream VPR benchmarks**. This provides strong evidence for the effectiveness of our proposed dynamic adaptive framework.
>
> To ensure these contributions are more prominent in the final version of the paper, we plan to make corresponding adjustments to the introduction and abstract.
>
> Thank you once again for your review; your feedback has been invaluable in helping us improve the quality of our paper.
>
> **References**
> ```
> [1] Izquierdo, Sergio, and Javier Civera. Close, but not there: Boosting geographic distance sensitivity in visual place recognition. ECCV 2024.
> [2] Lu, Feng, et al. Cricavpr: Cross-image correlation-aware representation learning for visual place recognition. CVPR 2024.
> [3] Lu, Feng, et al. SuperVLAD: Compact and robust image descriptors for visual place recognition. NIPS 2024.
> [4] Jia, Menglin, et al. Visual prompt tuning. ECCV 2022.
> [5] Qiu, Qibo, et al. Emvp: Embracing visual foundation model for visual place recognition with centroid-free probing. NIPS 2024.
> ```

---

> ### Author Response · Authors · 2025-11-21
> **Response to Reviewer NcXJ (Part 2/3)**
>
> **W2&3. Justification for Technical Choices and Comparison with Alternatives**
>
> Thanks for your comments to help us improve the paper. You have raised a crucial point regarding our rationale for choosing the current technical solutions over other alternatives. Below, we elaborate on the design philosophy and trade-offs within the SAGE framework. Our core objective was to build a **synergistic, lightweight, and dynamic training framework** to precisely address existing bottlenecks in VPR.
>
> **1. Rationale for Choosing SoftP for Feature Interaction**
>
> The key to VPR is identifying the most discriminative landmark details, which can be diluted when all image patches are treated equally. Complex alternatives like full attention mechanisms or object detectors introduce significant overhead and task bias. Similarly, hard semantic segmentation can discard useful texture information. In contrast to these approaches, SoftP offers a lightweight solution.
>
> Through its data-driven residual weighting, SoftP **gently amplifies the importance of key features without disrupting the geometric structure** of the pretrained features or adding significant computational cost. This aligns perfectly with our Parameter-Efficient Fine-Tuning (PEFT) design principle. Furthermore, a visual comparison (**Fig. 5**) of importance heatmaps empirically supports this choice. While SoftP, SALAD [1], and CFP [2] all highlight prominent static landmarks, SoftP's heatmaps focus more on fine-grained, stable architectural features while effectively suppressing uninformative regions.
>
> **2. Rationale for Using the InteractHead**
>
> Methods like NetVLAD [3] and SALAD [1] focus on aggregating local features within a single image. To enable the model to learn common features across different images, we introduced the InteractHead. It employs a lightweight Transformer encoder, a standard and efficient architecture, to facilitate information exchange among descriptors within a batch. This is a mature paradigm validated in recent top-tier publications [4,5] that effectively **enhances feature robustness without introducing excessive complexity**.
>
> **3. Rationale for Dynamic Geo-Visual Graph Mining**
>
> A core challenge in VPR is distinguishing between locations that are "**visually similar but geographically distant**." An ideal sampling strategy must therefore be aware of both geographic and visual information. After constructing the geo-visual graph, our goal is to sample from "dense and mutually confusing subgraphs," since these represent the scenarios the model finds most challenging. This objective led us to adopt the Weighted Greedy Clique Expansion algorithm.
>
> Crucially, the definition of a "hard sample" is not static; it evolves as the model's capabilities improve during training. This is the fundamental reason we chose a dynamic, online mining strategy. By reconstructing the graph each epoch, it keeps the model focused on the most critical confusion areas at any given time, enabling precision learning.
>
> We hope this explanation clearly illustrates the design principles behind SAGE. Thank you again for prompting us to delve deeper and clarify these critical points.
>
> **References**
> ```
> [1] Izquierdo, Sergio, and Javier Civera. Optimal transport aggregation for visual place recognition. CVPR 2024.
> [2] Qiu, Qibo, et al. Emvp: Embracing visual foundation model for visual place recognition with centroid-free probing. NIPS 2024.
> [3] Arandjelovic, Relja, et al. NetVLAD: CNN architecture for weakly supervised place recognition. CVPR 2016.
> [4] Lu, Feng, et al. Cricavpr: Cross-image correlation-aware representation learning for visual place recognition. CVPR 2024.
> [5] Lu, Feng, et al. SuperVLAD: Compact and robust image descriptors for visual place recognition. NIPS 2024.
> ```

---

> ### Author Response · Authors · 2025-11-21
> **Response to Reviewer NcXJ (Part 3/3)**
>
> **W4. Limitations in Dynamic Environments**
>
> Thank you for pointing this out. Candidly discussing and analyzing the model's limitations is crucial for the integrity of our work.
>
> We posit that the fundamental reason for SAGE's performance degradation is that the model prioritizes static, permanent structures within the scene. As can be seen from the heatmaps in **Figures 5 and 9**, the SoftP module learns to focus on stable landmarks such as building facades and windows, while actively ignoring non-discriminative or transient objects like roads, the sky, and moving vehicles/pedestrians. In highly dynamic scenarios, such as heavy traffic, these key static landmarks can be heavily occluded by numerous temporary objects. This prevents the model from extracting sufficient discriminative features.
>
> To address this, we have incorporated a new **Figure 6 in the main text (line 486)**, which presents SAGE's qualitative failure modes. These failures typically occur under extreme conditions where reliable static features are absent, such as severe viewpoint shifts or heavy occlusion from dynamic objects.
>
> **W5. More Comparisons with the Latest SOTA Methods**
>
> To address this, we have conducted supplementary experiments and added an **extended SOTA comparison in Table 13 of the Appendix (line 962)**. This new table compares SAGE against very recent methods from top-tier venues, including VLAD-BuFF (ECCV 2024), EDTFormer (TCSVT 2025), and EffoVPR (ICLR 2025). The results are summarized below:
>
> | Method | SPED | MSLS-val | Nordland | Tokyo24/7 | Pitts250k-test |
> | :--- | :---: | :---: | :---: | :---: | :---: |
> | | **R@1/5/10** | **R@1/5/10** | **R@1/5/10** | **R@1/5/10** | **R@1/5/10** |
> | VLAD-BuFF (ECCV 2024) [1] | 91.4 / 95.9 / 96.9 | 91.8 / 96.0 / 96.2 | 85.1 / 93.8 / 96.0 | 96.2 / 98.7 / 99.4 | 95.5 / 98.5 / 99.2 |
> | EDTFormer (TCSVT 2025) [2] | 92.4 / 95.9 / 96.9 | 92.0 / 96.6 / 97.2 | 88.3 / 95.3 / 97.0 | 97.1 / 98.1 / 98.4 | 95.9 / 98.8 / 99.3 |
> | EffoVPR (ICLR 2025)$^{\dagger}$ [3] | 93.1 / 97.9 / 98.4 | 92.8 / 97.2 / 97.4 | 95.0 / - / - | **98.7** / 98.7 / 98.7 | - / - / - |
> | SAGE (Ours) | **98.9 / 99.7 / 100** | **94.5 / 97.4 / 97.8** | **96.0 / 98.9 / 99.4** | 97.5 / **99.1 / 99.4** | **98.4 / 99.4 / 99.7** |
>
> $^{\dagger}$Reranking method
>
> As the table clearly demonstrates, our single-stage SAGE method consistently outperforms these latest SOTA methods on the vast majority of benchmarks. While the two-stage EffoVPR achieves a slightly higher R@1 on Tokyo24/7, it requires additional computational overhead. This new comparison strongly validates SAGE's superior and efficient performance against the most advanced published work.
>
> Once again, we sincerely thank you for your constructive feedback, which has undoubtedly helped us to further improve the quality of our paper.
>
> **References**
> ```
> [1] Khaliq, Ahmad, et al. VLAD-BuFF: burst-aware fast feature aggregation for visual place recognition. ECCV 2024.
> [2] Jin, Tong, et al. EDTformer: An Efficient Decoder Transformer for Visual Place Recognition. TCSVT 2025.
> [3] Tzachor, Issar, et al. Effovpr: Effective foundation model utilization for visual place recognition. ICLR 2025.
> ```

---

### Official Review · Reviewer_oNFe · 2025-11-01

**Soundness:** 3
**Presentation:** 3
**Contribution:** 3
**Rating:** 6
**Confidence:** 5

**Summary:**

This paper proposes SAGE, a unified training framework for Visual Place Recognition that enhances spatial-visual discrimination through dynamic graph-based sample mining and lightweight feature enhancement modules. The method introduces Soft Probing for adaptive patch weighting and an online geo-visual graph to continuously update hard samples during training. Extensive experiments demonstrate that SAGE achieves state-of-the-art performance across eight challenging VPR benchmarks with high parameter efficiency, using a frozen DINOv2 backbone and parameter-efficient fine-tuning.

**Strengths:**

1. The experimental results are exceptionally strong, demonstrating state-of-the-art or highly competitive performance across a diverse set of eight benchmarks, including challenging datasets like SPED, Nordland, and AmsterTime that test robustness to seasonal, illumination, viewpoint, and long-term temporal changes.

2. The proposed dynamic geo-visual graph mining strategy is a principled and well-motivated approach to hard sample mining. By reconstructing the graph each epoch, it effectively addresses a key limitation of static sampling strategies and ensures the training focuses on currently relevant challenging examples.

3. The Soft Probing module and InteractHead are lightweight yet highly effective, enhancing local feature discriminability and cross-image coherence with a negligible number of added parameters, showcasing an excellent performance-to-cost design.

4. The framework is highly efficient, achieving its top-tier performance by freezing the DINOv2 backbone and employing parameter-efficient fine-tuning, resulting in a significantly lower number of trainable parameters compared to many existing methods, which is a significant practical advantage.

**Weaknesses:**

1. While the per-epoch graph reconstruction overhead is noted to be marginal, it could become a non-trivial computational bottleneck when scaling to extremely large-scale datasets, and a more in-depth scalability analysis would be beneficial.
2. This paper only provides results on the MSLS-val dataset and lacks results on the MSLS-challenge dataset. Since the labels for MSLS-challenge are not publicly available and online testing is required, results on this dataset would be more convincing.

**Questions:**

Please see weaknesses.

---

> ### Author Response · Authors · 2025-11-21
> **Response to Reviewer oNFe**
>
> Thanks for your positive recommendation and insightful comments. The following are Responses to the Questions.
>
> **W1. Scalability Analysis for Extremely Large-Scale Datasets**
>
> First, we would like to point out that SAGE's training was already conducted on large-scale data, specifically a combination of the MSLS and GSV-Cities datasets. To more precisely quantify the overhead of our online graph creation, we compared it with the cliquemining method, with each epoch mining 200 batches. The results are as follows:
>
> **Table A: Per-Epoch Online Graph Creation**
> | Method | Mining (min) | Train (min) | Total (min) |
> | :--- | :---: | :---: | :---: |
> | cliquemining | 4.3 | 25.1 | 29.4 |
> | SAGE (w/o GWS) | 6.1 | 28.4 | 34.5 |
> | SAGE (full) | 6.2 | 28.4 | 34.6 |
>
> The experimental data indicates that although SAGE's graph creation step (6.2 minutes) is slightly more time-consuming than that of cliquemining (4.3 minutes), the resulting total training time per epoch (34.6 minutes) remains within a highly efficient and perfectly acceptable range. Furthermore, SAGE's scalability is ensured by several key designs that effectively prevent computational bottlenecks as the dataset size grows:
>
> *   Sparsity Constraint. During graph creation, we enforce sparsity through an affinity threshold and a strict top-$k$ neighbor selection. This significantly reduces unnecessary pairwise computations.
>
> *   Batch-based Sampling to Reduce Time Complexity. We randomly sample $P$ sequences from the same city and construct subgraphs only on these sampled batches. This design reduces the time complexity of Online graph creation from $O(N^2)$, which depends on the total number of images ($N$), to $O(B \times k)$, which depends only on the batch size ($B$). The core computation for each epoch's graph creation is confined within these small batches, thus avoiding an exponential increase in time as the dataset grows.
>
> *   Parameter Efficiency to Balance Overhead. As you noted, SAGE employs a PEFT strategy by freezing the DINOv2 backbone, which significantly reduces the number of trainable parameters. This lightweight training approach also indirectly mitigates the computational burden of the graph creation step.
>
> **W2. Missing Results on the MSLS-challenge Dataset**
>
> We thank you for your valuable suggestion. As per your recommendation, we have evaluated our SAGE model on the MSLS-challenge dataset to further validate its generalization capability. We have benchmarked the results against current state-of-the-art methods, as shown in the table below:
>
> | Method | MSLS-val (R@1/5/10) | MSLS-challenge (R@1/5/10) |
> | :--- | :---: | :---: |
> | EigenPlaces | 89.1 / 93.8 / 95.0 | 67.4 / 77.1 / 81.7 |
> | SelaVPR | 90.8 / 96.4 / 97.2 | 73.5 / 87.5 / 90.6 |
> | CricaVPR | 90.0 / 95.4 / 96.4 | 69.0 / 82.1 / 85.7 |
> | SuperVLAD | 92.2 / 96.6 / 97.4 | 75.3 / 86.8 / 89.9 |
> | BoQ | 93.8 / 96.8 / 97.0 | 79.0 / 90.3 / 92.0 |
> | EMVP | 93.9 / 97.3 / 97.6 | 77.4 / 89.1 / 91.6 |
> | SALAD-CM | 94.2 / 97.2 / 97.4 | **82.7** / 91.2 / 92.7 |
> | **SAGE (Ours)** | **94.5** / **97.4** / **97.8** | 82.4 / **91.3** / **93.1** |
>
> As the table demonstrates, our SAGE model performs excellently on the highly challenging, blind-test MSLS-challenge benchmark.
> While its Recall@1 score is marginally lower than that of SALAD-CM (82.4 vs. 82.7), our method surpasses the current state-of-the-art on both Recall@5 (91.3 vs. 91.2) and Recall@10 (93.1 vs. 92.7). This clearly demonstrates SAGE's superior overall accuracy within the top retrieved candidates. Simultaneously, SAGE continues to achieve the best performance on the MSLS-val set.
>
> Thanks again for your valuable comments and suggestions. Please let us know if you have any further concerns. We'd be very glad to address them.

---

### Comment · Area_Chair_LeVb · 2025-11-27
**Rebuttal and Discussion Phase**

Dear Reviewers,

Thank you again for your time and effort in reviewing this paper. We are approaching the discussion deadline. I kindly ask you to review the rebuttal and continue the discussion so that we can reach a well-considered decision.

---

### Author Response · Authors · 2025-12-01
**General Response**

Dear Reviewers, AC, SAC, and PC,

We sincerely thank you for the precious time and insightful feedback, which have significantly strengthened our manuscript! Overall, we are encouraged that you find that:

*   SAGE introduces a dynamic approach to hard sample mining and geo-visual graph creation, **addressing limitations in previous methods**. (Reviewers oNFe, NcXJ, EQqS)
*   The proposed dynamic geo-visual graph mining strategy is a principled and well-motivated approach to hard sample mining. **The idea is elegant and well-justified**. (Reviewers oNFe, EQqS)
*   **Good results Thorough evaluation against other methods**. (Reviewers oNFe, NcXJ, AM9N, EQqS)
*   Demonstrates impressive efficiency, resulting in a significantly lower number of trainable parameters, **which is a significant practical advantage**. (Reviewers oNFe, EQqS)

To address the concerns raised by the reviewers, overall, we have conducted several additional experiments:

*   We added extra experiments on the MSLS-challenge and SF-small datasets, showing SAGE’s highly competitive performance across ten benchmarks. (Reviewers oNFe, AM9N)
*   We added an experiment comparing the time cost and performance of online and offline graph creation, showing that the training time is fully acceptable. (Reviewers oNFe, EQqS)
*   We added ablations comparing our offline graph creation with prior methods, which still shows clear advantages. We also performed a τ-threshold ablation on MSLS-val. (Reviewer AM9N)
*   We added Fig. 6 showing failures in dynamic environments and expanded Table 13 with additional comparisons to recent SOTA methods demonstrating leading performance. (Reviewer NcXJ)

We have also clarified the following key points:

*   We outline SAGE’s core idea, the motivation behind its dynamic geo-visual graph mining strategy, justify key design choices, and highlight its low parameter cost and high efficiency.
*   We corrected appendix dataset descriptions, added the Eynsham citation, and expanded Related Work with Parameter-Efficient Fine-Tuning (PEFT) approaches to improve rigor.
*   We added comparisons and descriptions of additional methods (e.g., EffoVPR) and datasets (e.g., SF-XL), and corrected debatable passages and inconsistent notation to improve readability.

Thanks again to the reviewers for their insightful comments and helpful suggestions. We have comprehensively responded to the reviewers’ feedback and revised the manuscript accordingly.

---

Best regards,

The Authors

---

### Meta-Review · Area_Chair_9yqe · 2026-01-06

**Summary:**

This paper proposes SAGE, a novel method for Visual Place Recognition that introduces a dynamic geo-visual graph mining strategy to address limitations in static hard sample mining. The method couples this with lightweight feature enhancement modules (SoftP and InteractHead) while leveraging a frozen DINOv2 backbone and parameter-efficient fine-tuning. The authors demonstrate strong empirical results across ten challenging benchmarks, achieving state-of-the-art or highly competitive performance with significantly fewer trainable parameters than many previous methods.

All reviewers agreed on the good performance of the proposed method. Most reviewers believed that this work provided a significant practical advantage or addressed some limitations in previous methods.

Reviewer AM9N pointed out several issues in the paper. The rebuttal provided comprehensive responses and additional experiments. The updated manuscript has addressed several concerns raised during review. After reading the review, rebuttal, and updated manuscript, I lean toward recommending acceptance.

**Reviewer Concerns:**

The primary concerns centered on clarity of contributions, justification of design choices, and evaluation fairness. Reviewer NcXJ requested a clearer articulation of the key contributions and deeper analysis of component selection, while Reviewer AM9N raised issues regarding inaccurate descriptions, evaluation fairness, and the need for ablations (e.g., offline graph creation). Another concern raised by reviewer AM9N is that the description of the dataset contains hallucinations. The authors addressed these issues by clarifying the novelty of the dynamic paradigm, adding requested experiments (MSLS-challenge, SF-small, the ablation of offline graph creation, and hyperparameter analysis), and correcting dataset descriptions and citations.

**Reviewer Scores:**

The reviewers presented a divergence in scores. Reviewers oNFe (6) and EQqS (8) were positive, praising the strong results, elegant dynamic mining idea, parameter efficiency, and thorough evaluation. Reviewer NcXJ (4) was marginally negative, acknowledging improved performance but seeking greater clarity and justification. Reviewer AM9N (2) recommended rejection, primarily based on concerns about hallucinations, some claims, and experiments. The two sentences containing hallucinations were in Appendix A.3 used for the description of the previous dataset, which had a minor impact on the paper, and the authors have corrected them. Based on a comprehensive evaluation, I recommend that this paper be accepted.

---

### Decision · Program_Chairs · 2026-01-26

Accept (Poster)